# Second-Order Unsupervised Feature Selection via Knowledge Contrastive Distillation

## Abstract

Unsupervised feature selection aims to select a subset from the original features that are most useful for the downstream tasks without external guidance information. While most unsupervised feature selection methods focus on ranking features based on the intrinsic properties of data, they do not pay much attention to the relationships between features, which often leads to redundancy among the selected features. In this paper, we propose a two-stage Second-Order unsupervised Feature selection via knowledge contrastive disTillation (SOFT) model that incorporates the second-order covariance matrix with the first-order data matrix for unsupervised feature selection. In the first stage, we learn a sparse attention matrix that can represent second-order relations between features. In the second stage, we build a relational graph based on the learned attention matrix and perform graph segmentation for feature selection. Experimental results on 12 public datasets show that SOFT outperforms classical and recent state-of-the-art methods, which demonstrates the effectiveness of our proposed method.

## 1 Introduction

In the digital world, huge amounts of high-dimensional data (Guyon et al., 2004; Bajwa et al., 2016; Yang et al., 2019; Maoying et al., 2020) are captured every day. Due to the existence of irrelevant or redundant features, data in high dimensions may significantly increase the computational costs and bring challenges for efficient and effective data management. Dimensionality reduction is one of the most well-known techniques to address the above issue, which can be categorized into feature transformation and feature selection. Feature transformation, also known as representation learning, aims to project the original high-dimensional features into a new low-dimensional feature space. The new feature space is usually a linear or nonlinear combination of the original features and does not have semantic meanings, so it is hard to be interpreted. On the other hand, feature selection methods try to select a subset of relevant features from all available features based on a predefined criterion, maintaining physical meanings of the original features for better interpretability.

Without label information, unsupervised feature selection methods aim to select a feature subset that can preserve the intrinsic structure of the whole feature set accurately. There are many algorithms designed to solve the unsupervised feature selection problem. ReliefF (Robnik-Šikonja & Kononenko, 2003), HSIC (Gretton et al., 2005), Laplacian Score (He et al., 2005), SPEC (Zhao & Liu, 2007), SPFS (Zhao et al., 2011) evaluate features by their capability in preserving the pairwise sample similarity. UDFS (Yang et al., 2011), FSASL (Du & Shen, 2015), and TSFS (Mirzaei et al., 2020) employ pseudo labels as the supervision to guide the feature selection along with a sparse constraint. Most of these methods apply the linear feature selection matrices and select the representative features by ranking their feature weight vector. Such operations treat the feature set independently and fail to tackle the complex high-order relationship (Zhang et al., 2017; Zhu et al., 2020) among original features which inevitably brings in the redundancy among the selected features.

**Contributions**: We propose a two-stage Second-Order unsupervised Feature selection via knowledge contrastive disTillation (SOFT) model that incorporates the second-order covariance matrix with the first-order data. In the first stage, SOFT learns a sparse attention matrix to explore the second-order feature relationships. In the second stage, we perform graph segmentation on the learned attention matrix for feature selection. In summary, we highlight our contributions as follows: (1) We consider the second-order feature relationship in the unsupervised feature selection problem and propose

the SOFT algorithm to distill knowledge from both original features and their second-order feature covariance matrix; (2) Our SOFT learns a mask matrix on or off the covariance matrix and obtains the attention matrix and masked matrix. Throughout distilling the structural knowledge, the sparse attention matrix contains such knowledge as much as possible while excluding that from the masked matrix as well; (3) Different from selecting features according to weights, we propose a graph segmentation-based feature selection method on the attention matrix, where only one representative feature is selected from each segment to avoid redundancy; and (4) Experimental results validate that our SOFT model outperforms classical and recent state-of-the-art models on 12 public datasets. We also provide in-depth exploration for both stages to demonstrate the effectiveness of SOFT.

## 2 METHODOLOGY

**Motivation**. Unsupervised feature selection aims to select a small portion of the original features that are most useful for the downstream tasks without external guidance. Most previous methods focus on ranking features based on the values of individual features (He et al., 2005; Zhao & Liu, 2007; Yang et al., 2011) while neglect the high-order relationships between features. Unfortunately, this might lead to the redundancy of the selected features and further deteriorate downstream tasks. Figure 1 provides an illustrative example of selection results on *Sonar* (Rossi & Ahmed, 2015) and *Waveform* (Breiman et al., 1984) by Laplacian Score (LapScore) (He et al., 2005). The heatmap shows the initial relationships between features based on the covariance matrix of features. We remove the diagonal values and calculate the absolute values of covariance, which are normalized for visualization. In this example, we select the top four

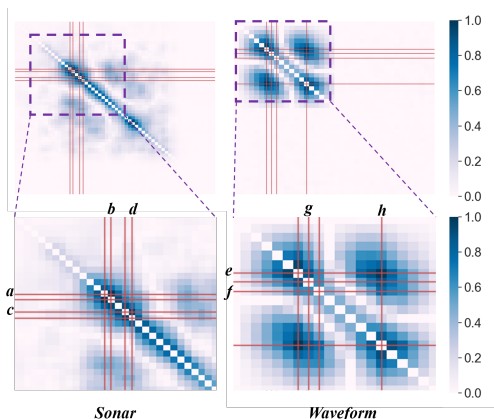

Figure 1: Visualization of feature relations of *Sonar* and *Waveform* datasets. Selected features by LapScore are marked by red lines.

features on *Sonar* and *Waveform* by LapScore, which are highlighted with red lines in Figure 1. Obviously, features $a$ and $b$, features $c$ and $d$ on *Sonar* are from two groups within of high similarity, indicating that one feature might be denoted by the other. Similarly, On *Waveform*, features $e$, $f$, and $h$ contain highly relevant information as well. However, features from the same feature group might lead to redundancy, which disobeys the purpose of feature selection. This drives us to explore the complex relationships derived from the second-order feature covariance matrix in the unsupervised feature selection problem. Motivated by the above situation, we design a method that can avoid such redundancy by considering both first-order data and second-order data.

**Framework Overview**. To incorporate the correlation feature information, we propose a two-stage model SOFT, which takes the first-order data matrix and second-order feature correlation matrix as inputs. In the first stage, SOFT learns a mask on the feature correlation matrix via knowledge contrastive distillation, which is beneficial to preserve the data structure. In the second stage, we select the features on the masked correlation matrix via graph segmentation.

Figure 2 shows the overview of our SOFT framework. With the first-order data matrix and second-order correlation feature matrix as inputs, we aim to learn a mask matrix applying to the feature correlation matrix for feature selection. By this means, we can obtain the attention matrix and masked matrix by the learnable mask on or off the feature matrix, respectively. The key idea of SOFT is to distill the structural knowledge via making the attention matrix contain such knowledge as much as possible while excluding that from the masked matrix as well. The green and red lines in Figure 2 demonstrate the knowledge contrastive distilling process. To achieve this, we adopt a shared Graph Convolution Network (GCN) (Kipf & Welling, 2016) to generate attention/original/masked representations from the attention/feature/masked matrices, respectively. Then we use pseudo labels generated by attention representation as positive guidance for original representation and negative guidance for masked representation so that attention representation and original representation are close to each other and far away from masked representation. Throughout the above knowledge contrastive distilling process, we can get a sparse and effective feature relation matrix that represent the second-order correlation, where each node denotes a feature and weights of edges are the

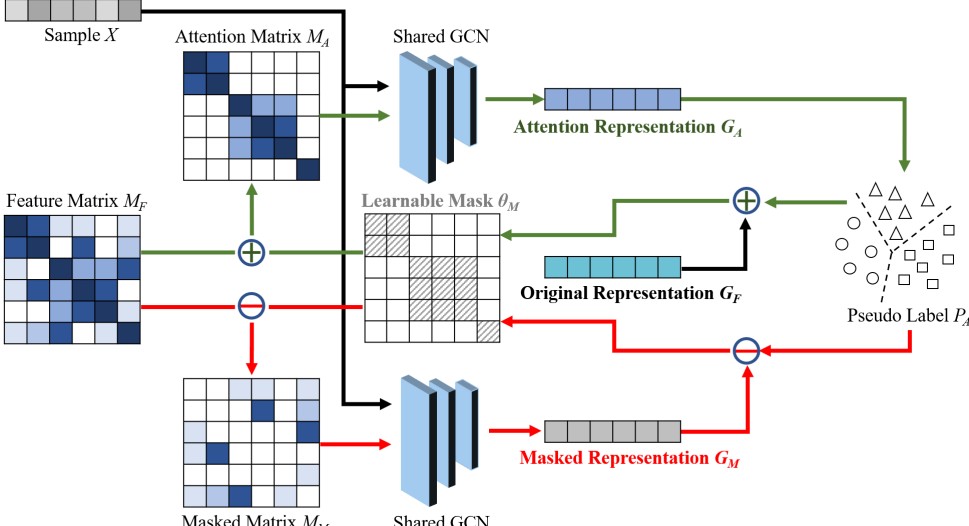

Figure 2: SOFT model framework. The inputs of our SOFT model consists of the first-order data matrix and second-order correlation feature matrix. The green line shows that attention representation is expected to preserve the intrinsic information from the pseudo labels and gets close to original representation, while the red line presents that the masked representation contains little structural information and gets away from the attention representation. Finally, we could get a sparse and effective feature relation matrix that can represent the second-order correlation.

corresponding values in the learned attention matrix. In the second stage, different from the existing work, which calculates the weight of each feature and suffers from the redundant feature issue (shown in Figure 1), our attention matrix delivers the weight for a pair of features. To proceed with the feature selection, we use graph segmentation to cut the attention matrix into partitions and select one feature that has the highest relationship to others from each partition as the final selection result. By this means, the redundancy among the selected features can be mitigated.

**Attention Matrix Learning**. Given $n$ data instances with $d$ features, we have the first-order $n \times d$ data matrix $X$ and the $d \times d$ second-order feature matrix $M_F = X^\top \times X$. Our SOFT model calculates a learnable mask for feature selection. In the first stage, SOFT consists of four components, attention layer, shared GCN, pseudo label generation, and contrastive learning. We use $\Theta = \{\theta_M, \theta_G, \theta_C\}$ to denote the trainable parameters set in the SOFT model, where $\theta_M$, $\theta_G$, and $\theta_C$ denote the parameters of the attention layer, shared GCN and contrastive learning, respectively. Note that the pseudo labels are generated from the attention

Table 1: Notations and description

| Notation | Dimension | Description |
|---|---|---|
| $n$ | scalar | Number of input samples |
| $d$ | scalar | Number of features |
| $c$ | scalar | Number of clusters |
| $X$ | $n \times d$ | Input data matrix |
| $M_F$ | $d \times d$ | Input feature matrix |
| $M_A$ | $d \times d$ | Attention matrix |
| $M_M$ | $d \times d$ | Masked matrix |
| $G_F$ | $n \times d$ | Representations generated by $X$ and $M_F$ |
| $G_A$ | $n \times d$ | Representations generated by $X$ and $M_A$ |
| $G_M$ | $n \times d$ | Representations generated by $X$ and $M_M$ |
| $P_F$ | $n \times c$ | Predictions of samples by $G_F$ |
| $P_A$ | $n \times c$ | Pseudo labels generated by $G_A$ |
| $P_M$ | $n \times c$ | Predictions of samples by $G_M$ |

matrix and shared GCN, which are controlled by $\theta_M$ and $\theta_G$. Therefore, no learnable parameters are needed for pseudo label generation. Table 1 shows the notations used in our SOFT model and their descriptions. Each part is detailed as follows.

*Attention Layer.* The initialized feature matrix $M_F$ may not be good enough to represent the relationship between features, so we add an attention layer to better capture second-order feature interactions by highlighting the important relations and reducing others through a learnable mask $\theta_M$. The attention matrix $M_A$, which represents the important part of $M_F$, is calculated as follows:

$$M_A = M_F \odot \theta_M, \tag{1}$$

where $\odot$ is the element-wise product. Due to the symmetry of $M_F$, $\theta_M$ is forced to be symmetric by adding a $d \times d$ parameter matrix with its transpose. With the learnable mask $\theta_M$ and the attention matrix $M_A$, We assume that the input feature matrix $M_F$ can be decomposed as a summation

formulation of attention matrix $M_A$ and masked matrix $M_M$. Then we define the masked matrix $M_M$ by the following function:

$$M_M = M_F - M_A. \tag{2}$$

To make the learned attention matrix $M_A$ sparse enough to identify crucial and principle relations, we apply $\ell_{2,1}$-norm to learnable mask $\theta_M$ on both row and column level, which is:

$$\mathcal{L}_{2,1} = \|\theta_M\|_{2,1} + \|\theta_M^\top\|_{2,1}. \tag{3}$$

*Shared Graph Convolutional Network.* GCN (Kipf & Welling, 2016) helps generate embeddings that contain both data information and feature relationship information. We apply a shared 2-layer GCN to extract the attention/original/masked representations from the attention/feature/masked matrices, respectively. In Kipf & Welling (2016), nodes of graphs denote samples, while in our case, nodes of graphs are features. Therefore, the computation of the GCN layer is a little different from Kipf & Welling (2016) and is described by the following equation:

$$G_{\theta_G}^{(l+1)} = \text{ReLU}(G_{\theta_G}^{(l)} \tilde{D}^{-\frac{1}{2}} \tilde{M} \tilde{D}^{-\frac{1}{2}} \theta_G^{(l)}), \tag{4}$$

where $M$ is the input relationship matrix, $\tilde{M} = M + I_d$, $I_d \in \mathbb{R}^{d \times d}$ is the identity matrix, $\tilde{D}$ is the degree matrix of $\tilde{M}$, and $\theta_G^{(l)}$ is a layer-specific trainable weight matrix. $G^{(l)}$ denotes the embeddings in the $l$-th layer, and $G^{(0)} = X$. In our case, we use a shared 2-layer GCN to process samples with different input relationship matrices $M_A$, $M_F$, and $M_M$. Thus we have the attention representation $G_A$, original representation $G_F$, and masked representation $G_M$, respectively.

*Pseudo Label Generation.* In the unsupervised feature selection, pseudo labels are usually used as the criterion to guide feature selection. In our experiments, we use Principal Component Analysis (PCA) (Wold et al., 1987) and K-means to generate pseudo labels (Caron et al., 2018). The embeddings processed by the clustering part is generated by attention matrix $M_A$, which is also the matrix we expect to learn in the SOFT model. PCA helps get principal features, and K-means clustering generates pseudo labels $P_A$ for the input samples $X$.

*Contrastive Learning.* With the above representations, we expect that attention representation and original representation are close to each other but far away from masked representation. To achieve such contrastive learning, we design two losses to measure the predictive abilities of different representations and compare them with the pseudo labels. Specially, we employ a 2-layer Multi-Layer Perceptron (MLP) to get predictions of the samples with original representation $G_F$ and masked representation $G_M$ about which clusters they belong to,

$$P_{\theta_C}^{(l+1)} = \sigma(P_{\theta_C}^{(l)} \times \theta_C^{(l)}), \tag{5}$$

where $\theta_C^{(l)}$ is a layer-specific trainable weight matrix, $P^{(l)}$ denotes the embeddings in the $l$-th layer, and $P^{(0)} = G_F$ or $P^{(0)} = G_M$ for the original representation and masked representation, respectively. We use $\sigma(\cdot) = \text{ReLU}(\cdot)$ as the activation function for layers before the last layer, and $\sigma(\cdot) = \text{Softmax}(\cdot)$ for the last layer. In our experiments, we adopt a shared 2-layer MLP and obtain $P_F$ and $P_M$ for original representation $G_F$ and masked representation $G_M$.

Pseudo labels $P_A$ generated by the clustering part are used for the following positive and negative training. By positive guidance for $P_F$, we get a cross-entropy loss $\mathcal{L}_F$ for $P_F$, which is defined as:

$$\mathcal{L}_F = -\sum_{i=1}^n \sum_{j=1}^c (P_{Aij} \log P_{Fij} + (1 - P_{Aij}) \log(1 - P_{Fij})), \tag{6}$$

where $P_{Fij}$ denotes the probability that the $i$-th sample belongs to cluster $j$ based on $M_F$. $P_{Aij} = 1$ if the $i$-th sample belongs to cluster $j$ based on the clustering result, otherwise $P_{Aij} = 0$. Eq. (7) is designed to make $P_F$ similar to pseudo labels $P_A$. While by negative guidance for predictions of the masked part $P_M$, we apply attention loss described in Li et al. (2018b) to reduce the weights of unimportant relations, which is stated as follows based on our notations:

$$\mathcal{L}_M = \sum_{i=1}^n \sum_{j=1}^c P_{Aij} P_{Mij}, \tag{7}$$

where $P_{Mij}$ denotes the probability that the $i$-th instance belongs to cluster $j$ based on $M_M$. Through minimizing Eq. (7), the model tends to put instances to a cluster that they do not belong to, thus driving predictions $P_M$ generated by masked representation different from pseudo labels $P_A$, which would finally lead to the difference of masked representation $G_M$ and attention representation $G_A$.

*Objective Function*. Combining Eq. (3), Eq. (6), and Eq. (7), Our overall objective function of SOFT can be written as follows:

$$\min_\Theta \mathcal{L}_F + \alpha\mathcal{L}_M + \beta\mathcal{L}_{2,1}, \tag{8}$$

where $\alpha$ and $\beta$ are hyperparameters for $\mathcal{L}_M$ and $\mathcal{L}_{2,1}$, respectively. We adopt Adam optimizer (Kingma & Ba, 2014) to minimize the objective function. The detailed algorithm can be found in Appendix A.

**Feature Selection on Attention Matrix**. Different from the traditional feature selection methods that return a weight vector to choose the top-ranked features, here, our attention matrix provides the weight of a pair of features. To proceed with the feature selection, we build a feature graph based on the attention matrix. Nodes of the graph denote features, and edges are the relationship between features in the attention matrix. Then we perform graph segmentation for feature selection.

*Graph Construction*. Instead of transforming the attention matrix $M_A$ to a graph immediately, we add two additional processes so that the generated graph is more suitable for graph segmentation and feature selection. The first one is to remove some irrelevant features. We calculate a score for each feature by $S = \sum_{i=1}^{d} M_{Ai}$, where $S$ is the summation of each row or column of the attention matrix, which measures the importance of each feature. We delete the last 10% features based on the sorted $S$. The second step is to set all values smaller than the median value in $M_A$ to zero so that the number of unimportant edges is largely reduced. We also set all negative values in $M_A$ to zero to make sure no negative edges would exist in the constructed graph.

*Graph Segmentation*. With the processed graph, we apply the graph segmentation method provided by Karypis & Kumar (1998) and cut the graph into $k$ parts, where $k$ is the number of features we aim to select. For each partition, we choose the feature that has the largest value in $S$, since we consider features in the same partition are highly related.

Through learning attention matrix in the first stage of our method, we incorporate both first-order data and second-order data and get a refined feature relation matrix that can better reflect feature relations. Then in the second stage, we build a graph based on the learned attention matrix and perform graph segmentation, which groups high-correlated features together. Therefore, by selecting features from each partition, our method reduces redundancies among the selected features. Our proposed method focuses on learning pair-wise relationships of features and uses graph segmentation to select features. The time complexity of our method is $O(nd^2)$. If the number of samples is larger than the number of features ($n > d$), the space complexity of our method is $O(nd)$, otherwise $O(d^2)$.

## 3 EXPERIMENTAL RESULTS

**Experimental Settings**. We introduce the experimental settings below.

*Datasets*. We select 12 public feature selection benchmark datasets of different types for evaluation: *COIL20* (Nene et al., 1996), *Colon* (Alon et al., 1999), *Gisette* (Guyon et al., 2004), *Lung-Cancer* (*L.Cancer*) (Hong & Yang, 1991), *Madelon* (Guyon et al., 2007), *MovementLibras* (*M.Libras*) (Dias et al., 2009), *NCI9* (Ross et al., 2000), *ORL* (Cai et al., 2006), *Sonar* (Rossi & Ahmed, 2015), *UAV1* and *UAV2* (Zhao et al., 2018), and *Waveform* (*Wave.*) (Breiman et al., 1984). The instance numbers of these dataset range from 32 to 19937, the feature numbers range from 40 to 9712, and the true cluster numbers are from 2 to 40. The sample/feature ratios of these datasets range from 0.01 to more than 350, indicating the diversity of the datasets. Detailed dataset statistics can be found in Appendix B.

*Comparative Methods and Implementation*. We choose 10 classical and recent state-of-the-art unsupervised feature selection methods for comparison. Laplacian Score (LapScore) (He et al., 2005) selects features by scoring features with a Gaussian Laplacian matrix. SPEC (Zhao & Liu, 2007) is a more general framework for feature selection based on spectral graph theory, where LapScore is a special case of it. MCFS (Cai et al., 2010) uses spectral analysis and sparse regression to select features and capture the multi-cluster data structure. UDFS (Yang et al., 2011) selects features by discriminative analysis and $\ell_{2,1}$ minimization. NDFS (Li et al., 2012) selects the most

Table 2: Results of different UFS methods on 10% selected features in terms of accuracy

| Dataset | LapScore | SPEC | MCFS | UDFS | NDFS | LRPFS | NSSLFS | TSFS | CAE | InfFS | SOFT |
|---|---|---|---|---|---|---|---|---|---|---|---|
| *COIL20* | 0.56±0.03 | 0.59±0.02 | 0.63±0.02 | 0.55±0.03 | 0.61±0.02 | 0.57±0.02 | 0.64±0.04 | 0.61±0.04 | 0.65±0.02 | 0.58±0.05 | **0.67**±0.01 |
| *Colon* | 0.54±0.01 | 0.55±0.00 | 0.53±0.00 | 0.52±0.00 | 0.55±0.00 | 0.55±0.01 | 0.55±0.00 | 0.54±0.03 | 0.53±0.01 | 0.55±0.07 | **0.56**±0.00 |
| *Gisette* | 0.67±0.00 | 0.70±0.00 | 0.80±0.00 | 0.58±0.00 | 0.74±0.01 | N/A⁺ | N/A⁺ | 0.57±0.00 | 0.62±0.00 | 0.61±0.04 | **0.81**±0.00 |
| *L.Cancer* | 0.75±0.00 | 0.59±0.00 | 0.69±0.00 | 0.65±0.01 | 0.69±0.00 | **0.78**±0.00 | **0.78**±0.00 | 0.60±0.05 | 0.56±0.02 | 0.69±0.10 | **0.78**±0.01 |
| *Madelon* | 0.51±0.00 | 0.51±0.00 | 0.53±0.00 | 0.51±0.00 | **0.60**±0.00 | 0.50±0.00 | 0.53±0.00 | 0.57±0.03 | 0.56±0.00 | 0.52±0.03 | **0.60**±0.00 |
| *M.Libras* | 0.27±0.01 | 0.29±0.01 | 0.38±0.01 | 0.35±0.02 | 0.43±0.01 | 0.39±0.01 | 0.37±0.01 | 0.36±0.01 | 0.43±0.02 | 0.44±0.03 | **0.46**±0.00 |
| *NCI9* | 0.44±0.02 | 0.43±0.02 | 0.38±0.02 | 0.44±0.03 | 0.43±0.02 | 0.42±0.03 | 0.37±0.03 | 0.44±0.02 | **0.46**±0.04 | 0.40±0.03 | 0.44±0.03 |
| *ORL* | 0.49±0.02 | 0.56±0.02 | 0.56±0.02 | 0.47±0.02 | 0.57±0.02 | 0.43±0.02 | 0.56±0.03 | 0.57±0.01 | 0.51±0.01 | 0.52±0.02 | **0.58**±0.01 |
| *Sonar* | 0.57±0.00 | 0.54±0.00 | 0.58±0.00 | 0.54±0.00 | 0.52±0.00 | **0.64**±0.00 | 0.60±0.00 | 0.56±0.04 | 0.56±0.00 | 0.58±0.00 | **0.64**±0.00 |
| *UAV1* | 0.56±0.00 | 0.67±0.00 | 0.54±0.00 | 0.55±0.00 | 0.65±0.00 | N/A⁺ | 0.60±0.00 | 0.65±0.01 | 0.56±0.00 | 0.55±0.00 | **0.78**±0.01 |
| *UAV2* | 0.80±0.00 | 0.60±0.00 | 0.76±0.00 | 0.81±0.00 | 0.58±0.00 | 0.55±0.00 | 0.56±0.00 | 0.64±0.01 | 0.81±0.00 | 0.58±0.02 | **0.82**±0.00 |
| *Wave.* | 0.51±0.00 | 0.34±0.00 | 0.51±0.00 | 0.52±0.00 | 0.49±0.00 | 0.48±0.00 | 0.37±0.00 | 0.52±0.00 | 0.51±0.00 | 0.51±0.00 | **0.54**±0.00 |
| Average | 0.55 | 0.53 | 0.57 | 0.54 | 0.57 | 0.52 | 0.54 | 0.55 | 0.56 | 0.54 | **0.64** |

N/A: "*" indicates the algorithmic running error, and "⁺" means the out-of-64GB-memory error.

Figure 3: Performance of 12 UFS methods on different percents of selected features. On each dataset, only top 5 methods on average performance are displayed for better visualization.

discriminative features with a nonnegative constraint and $\ell_{2,1}$ regularization. LRPFS (Zheng et al., 2018) adopts a low-rank constraint to preserve the subspace structure information. NSSLFS (Zheng et al., 2019) learns the feature weight matrix with the $\ell_{2,1}$-norm and the non-negative constraint based on the low-dimensional sparse subspace learning. TSFS (Mirzaei et al., 2020) employs a teacher-student scheme for deep feature selection. CAE (Balın et al., 2019) uses the concrete distribution and the reparametrization trick to differentiate through an reconstruction loss and select input features. InfFS (Roffo et al., 2020) is a fast graph-based approach which ranks and selects features by considering the possible subsets of features as paths on a graph.

For LapScore, SPEC, MCFS, UDFS, and NDFS, we adopt implementations and default settings provided by scikit-feature (Li et al., 2018a). For LRPFS and NSSLFS, we set the values of hyperparameters in their objective functions to 1.0. For TSFS, CAE (non-linear version), and InfFS, we use default settings provided in their open-source codes. The settings of our SOFT model are as follows.

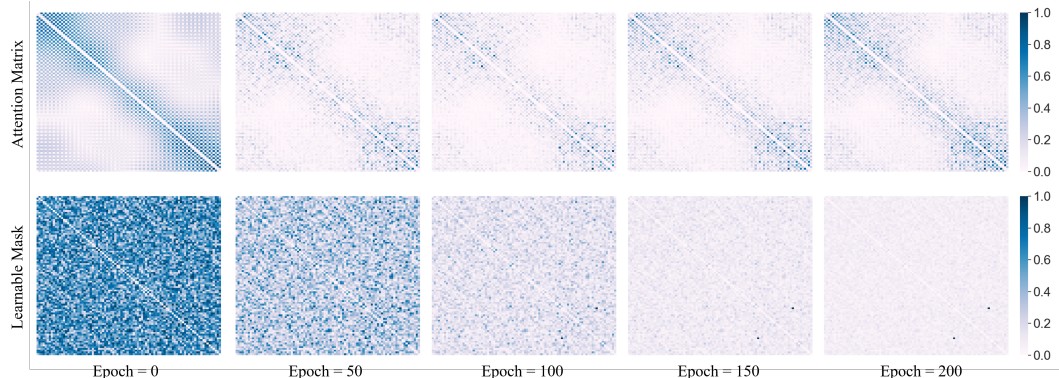

Figure 4: Visualization of attention matrix and learnable mask during iterations on *MovementLibras*.

In the stage of Attention Matrix learning, we implement the networks by PyTorch. The Learnable Mask is initialized by the normal distribution. The weights $\alpha$ and $\beta$ in the objective function are 1 and 0.001, respectively. We adopt adam optimizer (Kingma & Ba, 2014) to minimize the objective function and set the learning rate to 1e-4. We run a total of 300 epochs to learn the attention matrix. In the stage of graph segmentation, we first remove 10% features as irrelevances based on the learned attention matrix and set the least 50% values of the remaining 90% features in attention matrix to zero before graph construction. Then we perform graph segmentation and select 1 feature from each partition. For the re-implementation, the link of code and datasets can be found in Appendix C.

*Evaluation Metric*. We employ k-means++ (Arthur & Vassilvitskii, 2006) on the selected features, and compare the obtained partition and ground truth by clustering accuracy, settings of which follows scikit-feature (Li et al., 2018a).

**Algorithmic Performance**. Table 2 shows the experimental results of different unsupervised feature selection methods on 10% selected features in terms of accuracy. The best results are highlighted in bold. "N/A" means the corresponding method cannot process the dataset successfully due to running error or out of memory error. We can see that our SOFT model achieves the best on 11 of the 12 datasets. One of the possible reasons is that SOFT explores the second-order relationships among features, while other competitive methods only used the first-order data, while we incorporated second-order data in the feature selection process. By grouping features based on the learned attention matrix, our method can avoid the redundancy described in Section 2. Moreover, the average accuracy of SOFT is significantly better than other methods with large margins of 5% to 10%, which demonstrates the positive effects of second-order feature exploration on the unsupervised feature selection problem. Beyond the algorithmic effectiveness, we also provide the algorithmic execution time of different algorithms in Appendix D.

Next, we show the performance of all the methods on selecting different percents of features in Figure 3. LapScore achieves the best performance with 20% selected features on *Madelon*, LRPFS is always the best among these methods on *L.Cancer*, NSSLFS outperforms other methods with 15% and 20% selected features on *Colon*, and TSFS with 15% selected features is ranked the first on *NCI9*. It is worthy to note that sometimes the performance of SOFT is not so stable as other methods, such as on *L.Cancer* when the number of selected features increased. This is because the comparison methods generate a ranking list for features and select features based on the feature ranking scores, while SOFT select features based on the graph segmentation result. With the number of selected features increasing, the feature partitions might change notably, which results in a different selection. In general, SOFT delivers the compromising features with different percents compared with others, especially when the percent of selected features is relatively small. For results with all percentages and datasets, SOFA outperforms all other methods with an average accuracy of 61%, which is significantly better than two recent methods (CAE achieves 54%, and InfFS gets 51%).

**In-depth Exploration of SOFT**. We provide the in-depth exploration of SOFT from the following aspects. Additional experimental results can be found in Appendix E-H.

*Parameter Analysis*. There are two parameters in the objective function of SOFT, $\alpha$ and $\beta$, denoting the weight of attention loss and $\ell_{2,1}$-norm loss, respectively. We vary $\alpha$ and $\beta$ from $1e-3$ to $1e+3$ to explore the impact of these two parameters on the final performance. Figure 5(a) shows the results

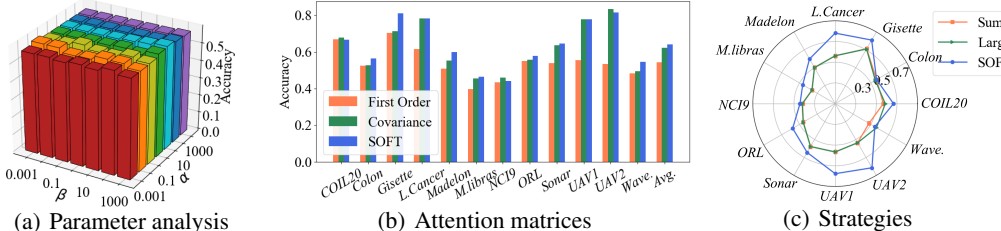

(a) Parameter analysis      (b) Attention matrices      (c) Strategies

Figure 5: In-depth exploration of SOFT. (a) Parameter analysis of $\alpha$ and $\beta$ on *Colon*, (b-c) performance of feature selection with different attention matrices and different feature selection strategies.

on *Colon* with 10% selected features. We can see that despite the large range of parameter values, the final performance does not change much, indicating that the Learnable Mask is well learned and SOFT might not be sensitive to the value of $\alpha$ and $\beta$ in a large range.

*Visualization of Attention Matrix and Learnable Mask.* To further analyze the performance of SOFT, we visualize the Attention Matrix and the corresponding Learnable Mask in different epochs on *MovementLibras*, which is shown in Figure 4. The darker color indicates the stronger correlation of the corresponding pair of features. To better recognize feature relations, we remove the diagonal values for visualization. In the beginning, there is no identified pattern in the Learnable Mask because the learnable mask is randomly initialized. As the training epochs increased, the learnable mask becomes sparser and seeks the dataset-dependent patterns. While the attention matrix is generated by an element-wise product of the original feature matrix and the learnable mask, the attention matrix became sparser as well. Therefore, the most important part of the attention matrix for representing samples was highlighted through network training.

Then we do a further step to explore the effectiveness of the learned attention matrix by comparing our results with two designated baseline methods. The first one is the First Order method, which only uses intrinsic properties of the data. We remove the GCN part of SOFT and utilizes a vectorial learnable mask for the first order network. The First Order learns the ranking scores for features, which is denoted by the Learnable Mask. The second one is the Covariance method, which uses the original feature matrix directly for the second stage of SOFT. Figure 5(b) shows the experimental result in terms of accuracy. On almost all the datasets, both SOFT and Covariance performed better than First Order, which demonstrates the effectiveness of incorporating feature relation in feature selection. While sometimes Covariance achieves a better result than SOFT, SOFT performed the best in most cases and achieved the best on average, proving that SOFT could learn a sparse and effective feature relation matrix to represent the second-order correlation.

*Feature Selection Strategy.* We adopt two baseline methods for feature selection from Attention Matrix as the comparison with the graph segmentation method in SOFT. The first baseline method, Weight Sum, utilizes a row-sum method to get the total relation value of each feature to all other features as the ranking scores for features. Features with the highest scores are selected. The second baseline method, Largest Weight, each time finds the largest value in Attention Matrix and selects the corresponding feature pair until $k$ features are selected. Comparison results among Weight Sum, Largest Weight, and the graph segmentation in SOFT are shown in Figure 5(c). Overall, the graph segmentation method in SOFT has a significant advantage over the baseline methods. This results from that SOFT avoids selecting highly correlated features by graph segmentation, thus reducing redundancies and bringing in more complementary features for performance boosting.

## 4 RELATED WORK

We briefly review related work on unsupervised feature selection and deep feature selection below.

**Unsupervised Feature Selection**. In the past decades, a large amount of unlabeled data are generated. To solve the feature selection problem for the unlabeled data, researchers have proposed many unsupervised feature selection methods (Liu et al., 2016; Li et al., 2016a), which can be divided into three main categories: Filter, Wrapper, and Hybrid. Filter methods evaluate features based on the data itself. Dash et al. (1997) propose one of the earliest filter unsupervised feature selection method, Sequential backward selection method for Unsupervised Data (SUD). SUD introduces a similarity matrix representing the pair-wise similarity between objects. By measuring the entropy

of the data, the relevance of each feature is quantified as a ranking score. Features with the highest scores are selected. Another example is Laplacian Score (He et al., 2005), which weights features according to their ability to preserve a predefined manifold structure represented by the Laplacian matrix. Similarly, SPECtrum decomposition (SPEC) (Zhao & Liu, 2007) also introduces an object similarity matrix. SPEC measures the consistencies between features and nontrivial eigenvectors of the Laplacian matrix and ranks features based on the consistencies. Wrapper methods select the most relevant features by using a clustering algorithm. (Dy & Brodley, 2004) introduce a method to select feature subsets using Expectation Maximization (EM) (Dempster et al., 1977) clustering and evaluate them with maximum likelihood and the scatter separability criterion. Kim et al. (2002) present an evolutionary multi-objective local selection algorithm to search feature subsets with K-means (Hartigan & Wong, 1979) and EM (Dempster et al., 1977) clustering. Instead of selecting feature subsets, Law et al. (2004) propose to estimate a set of real-valued quantities carried out by an EM algorithm through adopting a minimum message length (Wallace & Dowe, 2000) penalty. Hybrid methods try to take advantages of both approaches by adopting a two-stage process: filter stage and wrapper stage. In the filter stage, the features are scored based on the intrinsic properties of the data. And in the wrapper stage, feature sets are generated by a specific clustering algorithm. For instance, Dash & Liu (2000) adopt the method of Dash et al. (1997) for the filter stage to sort the features and the method of Dy & Brodley (2004) for the wrapper stage to build clusters. Solorio-Fernández et al. (2016) combine the spectral feature selection framework using the Laplacian Score (He et al., 2005) ranking and a modified Calinski–Harabasz index (Caliński & Harabasz, 1974). Different from the above-mentioned methods which are based on ranking, Kim & Gao (2006) propose a method that starts with the wrapper stage by Least-Square-Estimation based evaluation and then selects feature set through a Bayesian network in the filter stage. InfFS (Roffo et al., 2020) is a graph-based filtering approach, which evaluates the values of paths in a graph and selects discrete input features by exploiting properties of power series of matrices and the concept of absorbing Markov chains.

**Deep Feature Selection**. Recently, deep learning techniques have gained much attention and brought in some studies on deep feature selection (Chang et al., 2017; Taherkhani et al., 2018). DFS (Li et al., 2016b) adds a weight layer to Multi-layer Perceptron (MLP) together with a sparse regularization term so as to take advantage of deep structures to model nonlinearity. Zhao et al. (2015) propose to combine deep neural networks with sparse representation for grouped heterogeneous feature selection. The model first converts the multi-modal data into a unified representation, then selects features through solving a sparse group lasso (Friedman et al., 2010) problem. In recent years, some studies also involve data reconstruction error in deep unsupervised feature selection. AEFS (Han et al., 2018) jointly learns a self-representation autoencoder model and the importance weights of each feature for feature selection. Furthermore, GAFS (Feng & Duarte, 2018) not only adopts a single layer autoencoder but also incorporates the spectral graph analysis for learning. UDSFS (Cong et al., 2016) selects the most discriminative features and meanwhile designates appropriate weights to the feature dimensions by utilizing the group sparsity of the features. TSFS (Mirzaei et al., 2020) presents a teacher-student scheme for deep feature selection, in which a teacher network is used to learn low-dimensional representations, and a student network is employed for feature selection by minimizing the reconstruction error. CAE (Balın et al., 2019) uses a concrete selector layer as the encoder and a standard neural network as the decoder, stochastically selects discrete input features by concrete random variables and the reparametrization trick to get a subset of input features.

## 5 CONCLUSION

In this paper, we proposed a two-stage framework named SOFT for unsupervised feature selection, which incorporated second-order data with first-order data. Specifically, in the first stage, we first obtained the Attention Matrix and Masked Matrix by applying a learnable mask on and off the input Feature Matrix. Then we generated Attention/Original/Masked Representations from Attention/Feature/Masked Matrices by a shared Graph Convolutional Network. To train the learnable mask, we used pseudo labels generated by Attention Representation as positive guidance for Original Representation and negative guidance for Masked Representation, such that Attention Representation and Original Representation were similar to each other and different from Masked Representation. In the second stage, we constructed a graph based on the well-learned Attention Matrix and utilized graph segmentation to separate the graph into several parts. We chose one feature from each partition as the feature selection result. Experiments on public datasets demonstrated that our method outperformed classical and recent state-of-the-art methods on tackling the unsupervised feature selection problem.

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

# A ALGORITHM OF SOFT

---

**Algorithm 1** Second-Order unsupervised Feature selection via knowledge contrastive disTillation

---

**Input:** Input Data Matrix $X$; Feature Matrix $M_F = X^\top \times X$;
**Output:** $k$ selected features;
1: Initialize $\Theta = \{\theta_M, \theta_G, \theta_C\}$;
2: **repeat**
3:     Generate Attention Matrix $M_A$ and Masked Matrix $M_M$ by applying Learnable Mask on or off $M_F$;
4:     Use GCN to process $X$ with $M_A$, $M_F$ and $M_M$ to generate Attention Representation $G_A$, Original Representation $G_F$ and Masked Representation $G_M$;
5:     Adopt clustering on $G_A$ to get pseudo labels $P_A$;
6:     Employ MLP on Original Representation $G_F$ and Masked Representation $G_M$ to get predictions $P_F$ and $P_M$;
7:     Use $P_A$ as positive guidance for $P_F$, and negative guidance for $P_M$ to train the objective function;
8: **until** the model is converged.
9: Build a graph based on $M_A$ via removing noisy features;
10: Apply graph segmentation on the graph to get $k$ partitions and select one feature from each partition.

---

Algorithm 1 describes the whole process of SOFT. Through learning Attention Matrix in the first stage of our method, we incorporate both first-order data and second-order data and get a refined feature relation matrix that can better reflect feature relations. Then in the second stage, we build a graph based on the learned Attention Matrix and adopt the graph segmentation method, which groups high-correlated features together.

# B DATASET STATISTICS

We select 12 public feature selection benchmark datasets of different types for evaluation: *COIL20* (Nene et al., 1996), *Colon* (Alon et al., 1999), *Gisette* (Guyon et al., 2004), *Lung-Cancer* (*L.Cancer*) (Hong & Yang, 1991), *Madelon* (Guyon et al., 2007), *MovementLibras* (*M.Libras*) (Dias et al., 2009), *NCI9* (Ross et al., 2000), *ORL* (Cai et al., 2006), *Sonar* (Rossi & Ahmed, 2015), *UAV1* and *UAV2* (Zhao et al., 2018), and *Waveform* (*Wave.*) (Breiman et al., 1984). Tabel 3 shows the statistics of these datasets, where the sample/feature ratios range from 0.01 to more than 350, indicating the diversity of the datasets.

Table 3: Statistics of datasets

| Dataset | Type | #Sample | #Feature | #Cluster | Ratio | Density |
|---|---|---|---|---|---|---|
| *COIL20* | Face Image | 1440 | 1024 | 20 | 1.41 | 0.656 |
| *Colon* | Biological | 62 | 2000 | 2 | 0.03 | 0.584 |
| *Gisette* | Handwritten | 13500 | 5000 | 2 | 2.70 | 0.130 |
| *L.Cancer* | Biological | 32 | 56 | 2 | 0.57 | 0.940 |
| *Madelon* | Artificial | 2600 | 500 | 2 | 5.20 | 1.000 |
| *Mov.Libras* | Gesture | 360 | 90 | 15 | 4.00 | 1.000 |
| *NCI9* | Biological | 60 | 9712 | 9 | 0.01 | 0.503 |
| *ORL* | Face Image | 400 | 1024 | 40 | 0.39 | 1.000 |
| *Sonar* | Sonar Signal | 208 | 60 | 2 | 3.47 | 0.999 |
| *UAV1* | Traffic | 19380 | 54 | 2 | 358.89 | 0.972 |
| *UAV2* | Traffic | 17256 | 54 | 2 | 319.56 | 0.983 |
| *Wave.* | Artificial | 5000 | 40 | 3 | 125.00 | 0.997 |

# C RE-IMPLEMENTATION

Codes and datasets of our work can be found at https://github.com/ICLR2022submission/SOFT.

## D   TIME COST

We record the running time for each method on 10% selected features, which is detailed in Table 4. When the number of features is much larger than the number of instances, our method runs relatively slow but faster than methods employing the matrix inverse calculation like UDFS. But when there are much more samples than features, our method is faster than others except for LapScore, NSSLFS, and InfFS. This is because SOFT focuses on relations between features, where the number of features contributes more to the time cost than the number of instances. In general, the speed of SOFT is acceptable even for large-scale datasets.

Table 4: Time cost of experiments in seconds

| Dataset | LapScore | SPEC | MCFS | UDFS | NDFS | LRPFS | NSSLFS | TSFS | CAE | InfFS | SOFT |
|---|---|---|---|---|---|---|---|---|---|---|---|
| *COIL20* | 11.71 | 23.98 | 41.81 | 27.92 | 18.85 | 78.21 | 126.69 | 62.83 | 102.55 | 8.35 | 180.18 |
| *Colon* | 0.81 | 1.13 | 2.32 | 31.67 | 6.88 | 2.19 | 93.42 | 3.96 | 88.55 | 1.65 | 46.38 |
| *Gisette* | 52.65 | 1070.95 | 101.09 | 8024.12 | 2011.81 | N/A | N/A | 687.24 | 1599.45 | 529.53 | 15129.18 |
| *L.Cancer* | 0.57 | 0.87 | 0.69 | 0.85 | 0.63 | 0.03 | 0.17 | 2.79 | 138.78 | 0.10 | 1.86 |
| *Madelon* | 7.59 | 24.32 | 30.33 | 82.30 | 345.65 | 994.50 | 72.60 | 81.75 | 293.36 | 3.40 | 75.84 |
| *M.Libras* | 3.61 | 4.38 | 4.28 | 4.23 | 3.29 | 1.82 | 1.05 | 16.19 | 256.30 | 0.23 | 3.80 |
| *NCI9* | 3.61 | 3.46 | 105.29 | 3664.82 | 527.26 | 41.98 | 1833.71 | 11.42 | 333.74 | 42.55 | 1034.50 |
| *ORL* | 10.59 | 13.22 | 26.42 | 16.28 | 13.14 | 5.22 | 55.67 | 31.41 | 376.19 | 1.78 | 55.54 |
| *Sonar* | 0.98 | 1.25 | 1.15 | 1.11 | 1.02 | 0.56 | 0.39 | 7.93 | 414.20 | 0.05 | 2.17 |
| *UAV1* | 51.38 | 871.47 | 5176.43 | 27726.59 | 13020.19 | N/A | 41.65 | 508.86 | 605.86 | 5.02 | 57.40 |
| *UAV2* | 42.62 | 545.45 | 3770.58 | 17091.52 | 10011.04 | 81824.85 | 43.82 | 445.54 | 660.85 | 2.72 | 51.52 |
| *Wave.* | 7.07 | 31.46 | 128.46 | 515.26 | 857.80 | 2168.23 | 8.28 | 136.66 | 748.83 | 1.10 | 15.26 |

Note: All experiments ran on a physical machine with Ubuntu 18.04, total memory of 64GB, AMD Ryzen Threadripper 2920X 12-Core Processor, and an NVIDIA GP102 GPU.

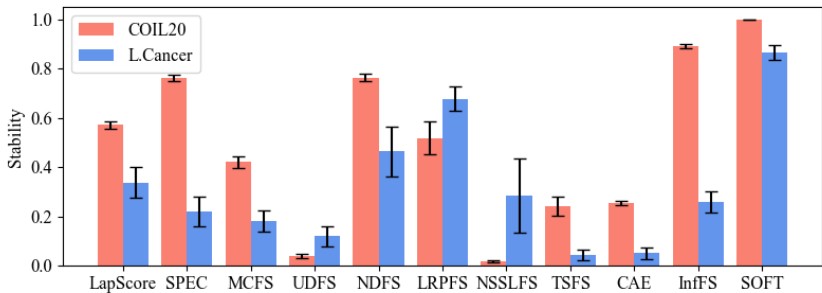

Figure 6: Stability of different methods with 10% selected features on *COIL20* and *L.Cancer*.

## E   STABILITY TEST

The "stability" of a feature selection algorithm refers to the robustness of its feature preferences, with respect to data sampling and to its stochastic nature. An algorithm is "unstable" if a small change in data leads to large changes in the chosen feature subset. A typical approach to measure stability is to first take $M$ bootstrap samples of the provided data set, apply feature selection to each one of them, and then measure the variability in the $M$ feature sets obtained. Here we conduct the stability tests on *COIL20* and *L.Cancer*. Specifically, we generate $M = 50$ bootstrap folds and run different unsupervised selection methods on these bootstrap folds. And then, we use the stability measurement proposed by Nogueira et al. (2017) ranging from -1 to 1, which returns the stability score with confidence intervals. The large value means more stable. The collection of the $M$ feature sets can therefore be modeled as a binary matrix $\mathcal{Z}$ of size $M \times d$, where $d$ is the dimension of features. A row in $\mathcal{Z}$ represents a feature set, and a column represents the selection of a given feature over the $M$ repeats as follows:

$$\mathcal{Z} = \begin{pmatrix} z_{1,1} & z_{1,2} & \cdots & z_{1,d} \\ z_{2,1} & z_{2,2} & \cdots & z_{2,d} \\ \vdots & \vdots & \ddots & \vdots \\ z_{M,1} & z_{M,2} & \cdots & z_{M,d} \end{pmatrix}. \tag{9}$$

Based on $\mathcal{Z}$, Nogueira et al. (2017) define the stability score as follows:

$$\hat{\Phi}(\mathcal{Z}) = 1 - \frac{\sum_{f=1}^{d} s_f^2}{\bar{k}(d - \bar{k})}, \text{ with}$$

$$\bar{k} = \frac{1}{M}\sum_{i=1}^{M}\sum_{f=1}^{d} z_{i,f}, \quad s_f^2 = \frac{M}{M-1}\hat{p}_f(1 - \hat{p}_f), \quad \hat{p}_f = \frac{1}{M}\sum_{i=1}^{M} z_{i,f}, \tag{10}$$

where $\bar{k}$ is the average number of features selected over the $M$ feature set, $s_f^2$ is the unbiased sample variance of the selection of the $f^{th}$ feature, and here $\hat{p}_f$ denotes the mean of the $f^{th}$ column of $\mathcal{Z}$.

Figure 6 shows the stability test of different feature selection methods on *COIL20* and *L. Cancer*. Our SOFT algorithm performs very stable with high scores and small intervals and excels others by a large margin in terms of stability scores.

# F   DIFFERENT WAYS TO GENERATE PSEUDO LABELS

SOFT is flexible with different pseudo label generation methods. So far, we use PCA + K-means to generate pseudo labels that are further used to guide feature selection. Here we test the performance of SOFT with other ways to generate pseudo labels in Table 5. Specifically, two popular deep clustering methods DEC (Xie et al., 2016) and IMSAT (Hu et al., 2017) with their default parameters are used here. We can see some improvements by the deep clustering methods. For example, SOFT with DEC and IMSAT outperforms our default setting (PCA + K-means) on *COIL20* and *NCI9*, which demonstrates the power of deep methods. But in general, the default setting is slightly better than the deep methods. We conjecture that some dedicated network architecture design is needed based on the data property to pursue better performance. If we compare the results in Table 5 and Table 2, SOFT with DEC and IMSAT to generate pseudo label still excels other baselines by a large margin on the average level. Moreover, we also report the execution time of SOFT with different ways to generate pseudo labels. Previously, SOFT might leave an impression of high computational cost just because of the long execution time on *Gisette* dataset. Actually, the majority of the running time is occupied by PCA. The speed of SOFT on *Gisette* dataset can be dramatically accelerated from 15129.18 to 739.55 or 1068.37 by second with DEC or IMSAT.

Table 5: Performance and execution time of SOFT with different ways to generate pseudo label

| Dataset | 10% selected features by accuracy | | | Execution time by second | | |
|---|---|---|---|---|---|---|
| | PCA+K-means | DEC | IMSAT | PCA+K-means | DEC | IMSAT |
| *COIL20* | 0.67±0.01 | 0.75±0.03 | 0.73±0.03 | 180.18 | 22.23 | 36.83 |
| *Colon* | 0.56±0.00 | 0.55±0.00 | 0.55±0.00 | 46.38 | 40.75 | 55.17 |
| *Gisette* | 0.81±0.00 | 0.73±0.00 | 0.76±0.00 | 15129.18 | 739.55 | 1068.37 |
| *L.Cancer* | 0.78±0.01 | 0.72±0.00 | 0.69±0.02 | 1.86 | 2.75 | 6.19 |
| *Madelon* | 0.60±0.00 | 0.56±0.00 | 0.56±0.00 | 75.84 | 18.25 | 35.79 |
| *M.Libras* | 0.46±0.00 | 0.46±0.02 | 0.49±0.02 | 3.80 | 4.17 | 8.03 |
| *NCI9* | 0.44±0.03 | 0.50±0.04 | 0.50±0.04 | 1034.50 | 6186.08 | 4550.16 |
| *ORL* | 0.58±0.01 | 0.63±0.02 | 0.58±0.02 | 55.54 | 15.06 | 23.86 |
| *Sonar* | 0.64±0.00 | 0.52±0.00 | 0.53±0.00 | 2.17 | 3.42 | 6.61 |
| *UAV1* | 0.78±0.01 | 0.67±0.00 | 0.66±0.00 | 57.40 | 94.37 | 166.51 |
| *UAV2* | 0.82±0.00 | 0.83±0.00 | 0.81±0.00 | 51.52 | 84.69 | 149.89 |
| *Wave.* | 0.54±0.00 | 0.55±0.00 | 0.51±0.00 | 15.26 | 25.21 | 44.79 |
| Average | 0.64 | 0.62 | 0.61 | 1387.80 | 603.04 | 512.68 |

# G   DIFFERENT GRAPH SEGMENTATION METHODS

Graph segmentation on the attention matrix is a crucial stage of SOFT. Here we change the graph segmentation algorithms with multilevel recursive bisection on different percents of selected features. Figure 7 shows our SOFT performance and standard deviations on different percents of selected features without and with multilevel recursive bisection. We can see that more smoothness over different percents is brought by multilevel recursive bisection on *L.Cancer* and *UAV2*, compared with the default one. The standard deviations on different percents of selected features are narrowed by multilevel recursive bisection as well.

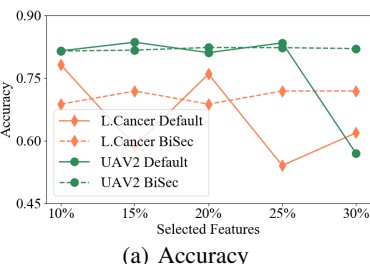
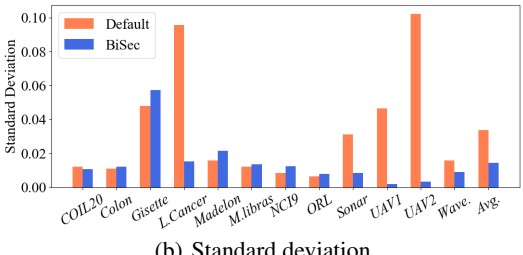

(a) Accuracy           (b) Standard deviation

Figure 7: SOFT performance and its standard deviation on different percents of selected features without and with multilevel recursive bisection. Default means without multilevel recursive bisection, and BiSec means with multilevel recursive bisection.

## H ABLATION STUDY ON NOISE REMOVAL FROM THE ATTENTION MATRIX

In the graph construction of SOFT, we have two ways to remove noises, including setting the values that are smaller than the median to be zero and deleting 10% features according to their importance. More details can be referred to in the paragraph of *Graph Construction*. Here we provide the ablation study on noise removal from the attention matrix in Table 6. Our default setting brings improvements on 7 out of 12 datasets compared with the single noise removal way and non-noise removal. On the average level, our default setting is also slightly better than others.

Table 6: Ablation study on noise removal from the attention matrix

| Dataset | Both (Our default) | Remove Values < Median | Delete 10% | No Noise Removal |
|---------|--------------------|-----------------------|------------|------------------|
| COIL20 | 0.67±0.01 | 0.71±0.03 | 0.73±0.03 | 0.74±0.03 |
| Colon | 0.56±0.00 | 0.55±0.01 | 0.56±0.01 | 0.55±0.00 |
| Gisette | 0.81±0.00 | 0.72±0.00 | 0.74±0.00 | 0.73±0.00 |
| L.Cancer | 0.78±0.01 | 0.72±0.04 | 0.69±0.00 | 0.72±0.04 |
| Madelon | 0.60±0.00 | 0.58±0.00 | 0.58±0.00 | 0.58±0.00 |
| M.Libras | 0.46±0.00 | 0.49±0.02 | 0.46±0.01 | 0.49±0.02 |
| NCI9 | 0.44±0.03 | 0.48±0.02 | 0.48±0.04 | 0.52±0.03 |
| ORL | 0.58±0.01 | 0.61±0.02 | 0.58±0.02 | 0.64±0.03 |
| Sonar | 0.64±0.00 | 0.53±0.00 | 0.50±0.00 | 0.52±0.00 |
| UAV1 | 0.78±0.01 | 0.78±0.00 | 0.78±0.00 | 0.78±0.00 |
| UAV2 | 0.82±0.00 | 0.81±0.00 | 0.81±0.00 | 0.81±0.00 |
| Wave. | 0.54±0.00 | 0.50±0.00 | 0.50±0.00 | 0.50±0.00 |
| Average | 0.64 | 0.62 | 0.62 | 0.63 |

## I ACKNOWLEDGEMENT

We would like to thank the anonymous reviewers for their suggestions and interactions during the review period.

