# OpenReview forum: "Second-Order Unsupervised Feature Selection via Knowledge Contrastive Distillation"
_ICLR.cc/2022/Conference — ICLR 2022 Submitted_

### Official Review · Reviewer_wcmh · 2021-10-31

**Correctness:** 3
**Technical Novelty And Significance:** 3
**Empirical Novelty And Significance:** 3
**Recommendation:** 6
**Confidence:** 4

**Main Review:**

Overall, I think it is an interesting approach to unsupervised feature selection. Although the computational cost, both spatial and temporal, is very high (a correlation matrix is used), the idea of searching for strong feature correlations is interesting. However, I have some concerns regarding the contribution:

1.- Some of the decisions are not clearly stated. For instance: In the graph construction, why the authors delete the las 10% features? Why they set to zero all values smaller than the median? In my opinion, the authors should include an ablation study that would shed some light over all those decisions.

2.- The same idea should be applied to the pseudo label generation and the evaluation metrics. I expect to see how this algorithm behaves when using a more complex approach than kmeans, like DEC [1] or IMSAT [2], for instance.

3.- Regarding to the problem of the accuracy drop whenever they increase the selected features (20% or higher), I wonder if there could be a different graph segmentation algorithm that could prevent it.


[1] Xie, J., Girshick, R., & Farhadi, A. (2016, June). Unsupervised deep embedding for clustering analysis. In International conference on machine learning (pp. 478-487). PMLR.

[2] Hu, W., Miyato, T., Tokui, S., Matsumoto, E., & Sugiyama, M. (2017, July). Learning discrete representations via information maximizing self-augmented training. In International conference on machine learning (pp. 1558-1567). PMLR.

**Summary Of The Paper:**

The authors propose a two-step unsupervised feature selection algorithm. Contraru to classic FS approaches, the authors try to infer important relations between pairs of features. In the final step, a graph segmentation approach is used to select the most important features. Experimental results show promising results.

**Summary Of The Review:**

Overall, I think it is an interesting idea with promising results, but more experiments have to be done to clearly state the performance of the algorithm, as well as a clear reasoning behing all the decisions the authors made in it.

---

> ### Author Response · Authors · 2021-11-17
> **Response to Reviewer wcmh (1/2)**
>
> Thanks for your comments, and we appreciate you recognizing the significance of our research problem and soundness of experiments. We are happy to provide extra experiments (Appendix E, F, G, H) and the following response and address all your concerns.
>
> ---
>
> **1. Ablation study on the ratio of removing features**
>
> This is a great suggestion that helps to understand our algorithm. We followed this suggestion and provided extra experiments with different noise removal strategies. Our default setting brings improvements on 7 out of 12 datasets compared with the single noise removal way and non-noise removal. On the average level, our default setting is also slightly better than others.
>
> Table. Results of our method with different noise removal strategies
>
> | Dataset  | Ours (Default) | Remove Values < Median | Delete 10% | No Noise Removal |
> |----------|----------------|------------------------|------------|------------------|
> | *COIL20*   | 0.67           | 0.71                   | 0.73       | 0.74             |
> | *Colon*    | 0.56           | 0.55                   | 0.56       | 0.55             |
> | *Gisette*  | 0.81           | 0.72                   | 0.74       | 0.73             |
> | *L.Cancer* | 0.78           | 0.72                   | 0.69       | 0.72             |
> | *Madelon*  | 0.60           | 0.58                   | 0.58       | 0.58             |
> | *M.Libras* | 0.46           | 0.49                   | 0.46       | 0.49             |
> | *NCI9*     | 0.44           | 0.48                   | 0.48       | 0.52             |
> | *ORL*      | 0.58           | 0.61                   | 0.58       | 0.64             |
> | *Sonar*    | 0.64           | 0.53                   | 0.50       | 0.52             |
> | *UAV1*     | 0.78           | 0.78                   | 0.78       | 0.78             |
> | *UAV2*     | 0.81           | 0.81                   | 0.81       | 0.81             |
> | *Wave.*    | 0.54           | 0.50                   | 0.50       | 0.50             |
> | Avg.     | 0.64           | 0.62                   | 0.62       | 0.63             |
>
> ---
>
> **2. Other methods to generate pseudo label**
>
> We followed the suggestion and replaced the PCA+K-means part in our algorithm with DEC [1] or IMSAT [2]. The results are demonstrated below. We can see that the deep clustering methods bring some improvements to some datasets. For example, SOFT with DEC and IMSAT outperforms our default setting (PCA + K-means) a lot on *COIL20* and *NCI9*, which demonstrates the power of deep methods. But in general, the default setting is slightly better than the deep methods. We conjecture that some dedicated network architecture design is needed based on the data property to pursue better performance. If we compare the results in the table below and other baseline methods (Table 2 in the manuscript), SOFT with DEC and IMSAT to generate pseudo labels still excels other baselines by a large margin on the average level.
>
> Table. Accuracy and time cost (by second) of our method with different pseudo labeling methods
>
> | Dataset  | Accuracy       |            |              | Time Cost      |            |              |
> |----------|----------------|------------|--------------|----------------|------------|--------------|
> |            | SOFT (Default) | SOFT (DEC) | SOFT (IMSAT) | SOFT (Default) | SOFT (DEC) | SOFT (IMSAT) |
> | *COIL20*   | 0.67           | 0.75       | 0.73         | 180.18         | 22.23      | 36.83        |
> | *Colon*    | 0.56           | 0.55       | 0.55         | 46.38          | 40.75      | 55.17        |
> | *Gisette*  | 0.81           | 0.73       | 0.76         | 15129.18       | 739.55     | 1068.37      |
> | *L.Cancer* | 0.78           | 0.72       | 0.69         | 1.86           | 2.75       | 6.19         |
> | *Madelon*  | 0.60           | 0.56       | 0.56         | 75.84          | 18.25      | 35.79        |
> | *M.Libras* | 0.46           | 0.46       | 0.49         | 3.80           | 4.17       | 8.03         |
> | *NCI9*     | 0.44           | 0.50       | 0.50         | 1034.50        | 6186.08    | 4550.16      |
> | *ORL*      | 0.58           | 0.63       | 0.58         | 55.54          | 15.06      | 23.86        |
> | *Sonar*    | 0.64           | 0.52       | 0.53         | 2.17           | 3.42       | 6.61         |
> | *UAV1*     | 0.78           | 0.67       | 0.66         | 57.40          | 94.37      | 166.51       |
> | *UAV2*     | 0.81           | 0.83       | 0.81         | 51.52          | 84.69      | 149.89       |
> | *Wave.*    | 0.54           | 0.55       | 0.51         | 15.26          | 25.21      | 44.79        |
> | Avg.     | 0.64           | 0.62       | 0.61         | 1387.80        | 603.04     | 512.68       |

---

> > ### Author Response · Authors · 2021-11-17
> > **Response to Reviewer wcmh (2/2)**
> >
> > ```
> > [1] Xie, J., Girshick, R., & Farhadi, A. (2016, June). Unsupervised deep embedding for clustering analysis. In International conference on machine learning (pp. 478-487). PMLR.
> > [2] Hu, W., Miyato, T., Tokui, S., Matsumoto, E., & Sugiyama, M. (2017, July). Learning discrete representations via information maximizing self-augmented training. In International conference on machine learning (pp. 1558-1567). PMLR.
> > ```
> >
> > ---
> >
> > **3. Different graph segmentation algorithm**
> >
> > Excellent suggestion! We also believe that the large performance changes on different percents of selected features come from the graph segmentation algorithm. Due to the non-inheritance of partitions with different cluster numbers, i.e., the numbers of selected features, the partition results might change a lot. Here we followed this suggestion and provided the extra experiments with different graph segmentation algorithms. The table below shows the standard deviations of SOFT on different percents of selected features without and with multilevel recursive bisection. We can see that more smoothness over different percents is brought by multilevel recursive bisection on *L.Cancer* and *UAV2*, compared with the default one. We invite the reviewer to refer to our new content, Figure 7 in Appendix G, for a better visual experience.
> >
> > Table. Standard deviation over different percents of SOFT default graph segmentation method and bi-segmentation method on 12 datasets
> >
> > | Dataset  | Ours(Default) | BiSeg  |
> > |----------|---------------|--------|
> > | *COIL20*   | 0.0120        | 0.0105 |
> > | *Colon*    | 0.0110        | 0.0121 |
> > | *Gisette*  | 0.0479        | 0.0572 |
> > | *L.Cancer* | 0.0957        | 0.0153 |
> > | *Madelon*  | 0.0159        | 0.0216 |
> > | *M.Libras* | 0.0121        | 0.0136 |
> > | *NCI9*     | 0.0085        | 0.0125 |
> > | *ORL*      | 0.0063        | 0.0078 |
> > | *Sonar*    | 0.0312        | 0.0083 |
> > | *UAV1*     | 0.0465        | 0.0018 |
> > | *UAV2*     | 0.1022        | 0.0033 |
> > | *Wave.*    | 0.0157        | 0.0089 |
> > | *Avg.*     | 0.0337        | 0.0144 |
> >
> > ---
> >
> > **4. Efficiency**
> >
> > Admittedly the calculation of the second-order feature matrix is time-consuming. When the number of features is much larger than the number of instances, our method runs relatively slow. We guess that SOFT might leave an impression of high computational cost just because of the long execution time on the *Gisette* dataset. Actually, the majority of running time was occupied by PCA. We followed the suggestion to try different ways to generate pseudo labels by deep clustering methods. We found the speed of SOFT on the *Gisette* dataset was dramatically accelerated from 15129.18 to 739.55 or 1068.37 by second with DEC [1] or IMSAT [2]. Please see the table in Q2 for the execution time.
> >
> > ```
> > [1] Xie, J., Girshick, R., & Farhadi, A. (2016, June). Unsupervised deep embedding for clustering analysis. In International conference on machine learning (pp. 478-487). PMLR.
> > [2] Hu, W., Miyato, T., Tokui, S., Matsumoto, E., & Sugiyama, M. (2017, July). Learning discrete representations via information maximizing self-augmented training. In International conference on machine learning (pp. 1558-1567). PMLR.
> > ```
> >
> > ---
> >
> > Thanks very much for these constructive suggestions. They are quite helpful in understanding our algorithm and delivering clear reasoning behind all the decisions within the algorithm. All the revised parts, as mentioned in the above point-to-point response, can be found in green in the updated paper. Further interactions are highly welcome, and we are happy to address your comments.

---

### Official Review · Reviewer_kEpS · 2021-11-02

**Correctness:** 3
**Technical Novelty And Significance:** 2
**Empirical Novelty And Significance:** 2
**Recommendation:** 3
**Confidence:** 5

**Main Review:**

For unsupervised feature selection, this paper proposes a two-stage second-order method. Knowledge contrastive distillation is also incorporated for feature learning. Experiments on various datasets validate the effectiveness.

Pros:

(1) The overall presentation and organization are good.

(2) The presented results are good on these related datasets.

Cons:

(1) My main concern lies in the experiments. NDFS achieves the best results among the compared baselines. However, NDFS is published in AAAI 2012, which is 9 years ago. The improvement of the proposed method over NDFS is also marginal on several datasets. Besides, the difference among these baselines is also marginal. Though the authors also compare with two recent methods, including CAE and InfFS, their performance is even lower than NDFS since they adopted different datasets in their original papers.  Therefore, the experimental results are not convincing.

(2) The novelty is not satisfying. All these modules, including the second-order information, have been well studied in the area of unsupervised learning. The proposed method is a combination of these methods. The perspective for unsupervised feature selection by feature relationship learning and graph segmentation is not new from my point of view.

(3) For the clustering task, many methods have already achieved much better results on COIL20, ORL, and even larger datasets, including CIFAR10 and CIFAR100.

(4) The computational complexity is very expensive, which is much higher than many compared methods.

(5) This paper is rejected by NeurIPS 2021, and I happened to review this paper several months ago. The authors do not incorporate any suggestions in this resubmitted manuscript. The main body is totally the same, and only one dataset for evaluation is changed.



**Summary Of The Paper:**

For unsupervised feature selection, this paper proposes a two-stage second-order method. Knowledge contrastive distillation is also incorporated for feature learning. Experiments on various datasets validate the effectiveness.



**Summary Of The Review:**

This paper proposes a rather complex and expensive new method for unsupervised feature selection that performs marginally better than the baselines (many quite old) on the set of datasets they have chosen to present in the paper. The exposition could also be clearer, but the idea seems more complex than it is interesting. In all, the paper does not rise to the standards of significance and novelty of ICLR.

---

> ### Author Response · Authors · 2021-11-17
> **Response to Reviewer kEpS (1/2)**
>
> Thank Reviewer kEpS for reviewing our paper twice. In our last submission, we provided a long and detailed response but failed to receive any feedback from you for further discussion or interaction. Hopefully, this time we can see your participation in the discussion so that we can either cancel some misunderstandings or get some specific suggestions.
>
> We are open and welcome to any comments that can further improve our paper. IMHO, however, these comments of the last round and this round from the reviewer are too general, which only contain the reviewer's opinion without supportive evidence. Without any specific suggestions, we are confused about how to improve our paper. Here we are not teaching the reviewer how to write a high-quality review in a professional and responsible fashion. If the reviewer likes it, feel free to read other reviewers’ comments. All of them either posted some clarification questions or provided constructive comments. We can feel from our heart that they truly helped us to improve our paper by providing suggestions and directions to tackle the disadvantages, rather than only criticizing our paper. Instead, we politely and sincerely post our response and expectation just from the authors’ perspective.
>
> ---
>
> **1. Experiments**
>
> NDFS, a classical unsupervised feature selection method, achieves the second-best results among all the competitive methods with 10% selected features. This is a fact. We are confused why the reviewer has a major concern on a fact and what the reviewer would like us to do. Our method outperforms NDFS by 7% on the average level. On  *L.Cancer*, *Sonar*, *UAV1*, and *UAV2* datasets, our method has 7%, 9%, 12%, and 24% improvements over NDFS. We would like to invite everyone to evaluate whether our improvements are margin or significant.
>
> In this version, we also added two recent methods CAE and InfFS, where their performance is even lower than NDFS (CAE and InfFS did not compare with NDFS in their original papers). Again, this is another fact. We are not surprised that some classical methods can achieve better performance than the recently proposed methods on some datasets. We all know that no method can win on all the datasets. Note that 12 datasets in our paper are representative and widely used in the unsupervised feature selection domain.
>
> ---
>
> **2. Novelty**
>
> The reviewer mentioned that “all these modules, including the second-order information, have been well studied in the area of unsupervised learning.” We sincerely solicit the reviewer to be more specific and provide some related papers. We are happy to learn from them.
>
> ---
>
> **3. Clustering Method**
>
> In the last round, the reviewer provided three deep clustering methods. Clearly, they are different problems from the unsupervised feature selection problem we address here. Please allow us to be bold and re-introduce the basic knowledge of the unsupervised feature selection problem, which helps address this concern on the clustering task. Unsupervised feature selection aims to select a subset of the original features in an unsupervised fashion. Since the labels are not available during the selection process, the cluster analysis technique as a downstream task is conducted on the selected features for the evaluation of the unsupervised feature selection task. Therefore, we focus on feature selection rather than cluster analysis (We definitely know many deep methods have already achieved much better results on *COIL20*, *ORL*, which seek a better representation for clustering, rather than the original features). In the literature of unsupervised feature selection, K-means clustering is widely used as a standard tool in this scenario due to its simplicity, popularity, and efficiency. Please check the literature on unsupervised feature selection. If necessary, we can provide some classical papers for the reviewer’s reference.
>
> ---

---

> > ### Author Response · Authors · 2021-11-17
> > **Response to Reviewer kEpS (2/2)**
> >
> > ---
> >
> > **4. Computational Complexity**
> >
> > Admittedly the calculation of the second-order feature matrix is time-consuming. When the number of features is much larger than the number of instances, our method runs relatively slow. We guess that SOFT might leave an impression of high computational cost just because of the execution time on the *Gisette* dataset. Actually, the majority of running time was occupied by PCA. We followed the suggestion from Reviewer wcmh to try different ways to generate pseudo labels by deep clustering methods. We found the speed of SOFT on the *Gisette* dataset was dramatically accelerated from 15129.18 to 739.55 or 1068.37 by second with DEC [1] or IMSAT [2]. More results can be found in the table below. We invite the reviewer to re-evaluate the computational cost of our algorithm.
> >
> > Table. Time cost by second of our method with different pseudo labeling methods
> >
> > | Dataset  | SOFT (Default) | SOFT(DEC) | SOFT(IMSAT) |
> > |----------|----------------|-----------|-------------|
> > | *COIL20*   | 180.18         | 22.23     | 36.83       |
> > | *Colon*    | 46.38          | 40.75     | 55.17       |
> > | *Gisette*  | 15129.18       | 739.55    | 1068.37     |
> > | *L.Cancer* | 1.86           | 2.75      | 6.19        |
> > | *Madelon*  | 75.84          | 18.25     | 35.79       |
> > | *M.Libras* | 3.80           | 4.17      | 8.03        |
> > | *NCI9*     | 1034.50        | 6186.08   | 4550.16     |
> > | *ORL*      | 55.54          | 15.06     | 23.86       |
> > | *Sonar*    | 2.17           | 3.42      | 6.61        |
> > | *UAV1*     | 57.40          | 94.37     | 166.51      |
> > | *UAV2*     | 51.52          | 84.69     | 149.89      |
> > | *Wave.*    | 15.26          | 25.21     | 44.79       |
> > | Avg.     | 1387.80        | 603.04    | 512.68      |
> >
> > ```
> > [1] Xie, J., Girshick, R., & Farhadi, A. (2016, June). Unsupervised deep embedding for clustering analysis. In International conference on machine learning (pp. 478-487). PMLR.
> > [2] Hu, W., Miyato, T., Tokui, S., Matsumoto, E., & Sugiyama, M. (2017, July). Learning discrete representations via information maximizing self-augmented training. In International conference on machine learning (pp. 1558-1567). PMLR.
> > ```
> >
> > ---
> >
> > **5. Second-time Review**
> >
> > We are open and welcome to any comments that can further improve our paper. By doing this, we can learn from the reviewers and increase our acceptance rate. In this revision, we added two recent competitive methods and replaced one dataset, which followed other NeurIPS reviewers’ comments. In our eyes (no offense), your comments are too general without any specific suggestions, which makes us hard to follow. We would like to see some concrete comments, for example, the relationship between our paper and another key reference, experimental comparison with another baseline method, or performance of our method on some datasets with some properties.
> >
> > ---
> >
> > **6. Other Issues**
> >
> > The reviewer summarized that “This paper proposes a rather complex and expensive new method for unsupervised feature selection that performs marginally better than the baselines (many quite old) on the set of datasets they have chosen to present in the paper.” (i) We are happy to hear from the reviewer any constructive comments to simplify our proposed method; (ii) We compared with 10 baseline methods, including 5 classical ones and 5 recent ones since 2018. As Reviewer 3WQu requested, we added two more recent methods, AEFS [3] and AGNOS [4], for comparison in the response (Please refer to Reviewer 3WQu’s section). ; (iii) 12 datasets in our paper are representative and widely used in the unsupervised feature selection domain.
> >
> > ```
> > [3] Han et al., Autoencoder inspired unsupervised feature selection. In International Conference on Acoustics, Speech and Signal Processing, 2018.
> > [4] Doquet et al., Agnostic feature selection. In Joint European Conference on Machine Learning and Knowledge Discovery in Databases, 2019.
> > ```
> >
> > ---
> >
> > We look forward to having the academic discussion in a professional and friendly manner :D~

---

> > ### Comment · Reviewer_kEpS · 2021-11-24
> > **Concerns about the paper**
> >
> > The reason why you do not receive further comments in the last round review is that no reviewer champions your paper after reading your response in the discussion. Since you think my original comments are too general, I give more details in the following about my concerns:
> >
> > (1) The experiments are not convincing. The authors claim that their experiments are sufficient since they adopt 12 datasets for evaluation. However, according to the statistics of datasets, most of these datasets have only two or three clusters (8 datasets). Among these 4 datasets with more than 3 clusters, the largest dataset only contains 1440 samples. From this perspective, the datasets adopted by this paper are not sufficient and representative. Even the Inf-FS adopt the more challenging CalTech 101 with 10K samples and 102 classes for verification.  More importantly, the authors claim that these datasets are commonly used for feature selection. However, I have a quick look at several compared papers, including NDFS, TSFS, CAE, InfFS, the adopted datasets are mostly different. Therefore, most of the reported results are based on reproduction, which might lead to unsatisfying results. On a commonly adopted dataset COIL20, the improvement is marginal. Therefore, I suggest the authors conduct more experiments on more challenging or commonly used datasets for a fair comparison instead of just adopting these datasets that your method is good on.
> >
> > (2) For the unsupervised feature selection task, the authors only reported the results on the clustering task. I think it is natural and reasonable to request the authors to compare with clustering-related methods. Otherwise, please give more comparison on other practical tasks. If the proposed method can only be used for clustering, since other methods already achieve much better results on this task, please give me a reason why I choose your method.
> >
> > (3) From my perspective, the contrastive learning-based unsupervised feature learning methods, such as SimCLR and MoCo, can also be regarded as feature selection since these methods take the high-dimensional image itself as input and output a low-dimensional feature representation for each image or sample. These methods can be applied to much larger datasets and can learn very discriminative feature representation for various downstream tasks, including classification, clustering, detection, segmentation, and so on. Many deep learning-based clustering methods also can be regarded as feature learning methods, such as DEC and DEPICT. The neural network in these methods has the same function as the projection matrix in feature selection. Therefore, I also think it is reasonable to compare with them.
> >
> > (4) The proposed method is really complicated. For attention matrix learning, this paper adopts GCN and contrastive learning, after which, the authors construct the graph based on artificially defined steps. Then a graph segmentation method is adopted for selection. The above process is also of high computational complexity. Though the authors presented additional processing time by incorporating deep clustering methods, it is still much higher than many compared methods, which limits its application for datasets with high-dimensional features. Besides, the deep clustering methods for pseudo label generation further make this paper more complicated. With more complicated processes and higher computational complexity, the improvement on the commonly used dataset COIL20 is still marginal, which is unacceptable.

---

> > > ### Author Response · Authors · 2021-11-25
> > > **Response to Reviewer kEpS's arrogance and unqualified expertise**
> > >
> > > It is nice to receive the feedback from Reviewer kEpS. But we sincerely feel sorry for Reviewer kEpS's arrogance and unqualified expertise.
> > >
> > > **Arrogance**
> > >
> > > Although this is beyond academic discussion, we would like to kindly point out that arrogance is not a good character for the professional review service.  Reviewer kEpS mentioned that "The reason why you do not receive further comments in the last round review is that no reviewer champions your paper." **This statement is logically unsound.** (Lack of logic is reflected not only on this sentence but also along with the whole comments; we will explain below) Response or not has no correlation with whether you champion our paper. In this round, you responded to us. But we believe you still do not support our paper. The logic of Reviewer kEpS is self-contradictory.
> > >
> > > **Unqualified Expertise**
> > >
> > > We tried to be euphemistic, but the comments from Reviewer kEpS exposed that the reviewer is definitely not an expert in the unsupervised feature selection domain (Although this statement is our conjecture, we are of high confidence. AC might help check the publication record of Reviewer kEpS. If our conjecture is correct, there should be no publication from Reviewer kEpS in the unsupervised feature selection domain). What's worse, the reviewer was reluctant to learn.
> > >
> > > **(1) Reviewer kEpS confounded unsupervised feature selection and clustering**
> > >
> > > We have clearly explained the difference between two concepts in our previous response. If the reviewer can take the patience and intend to learn more in this round, we are still happy to explain more. Unsupervised feature selection aims to select original features for the agnostic downstream tasks. However, there is no quantitive evaluation on the performance of selected features. Usually, a downstream task, such as clustering or classification, is employed as an evaluation tool for feature selection. In the unsupervised setting, we choose cluster analysis as the downstream task for evaluation. This is a professional standard in the unsupervised feature selection domain. Differently, the cluster analysis task is another story. The papers provided by Reviewer kEpS, including deep clustering methods and SSC, target to learn a better representation (*new* features) for clustering. In the unsupervised feature selection, only *original* features can be used for the downstream task, i.e., clustering.
> > >
> > > If Reviewer kEpS considers a clustering problem, a clustering method should be used. Here we target unsupervised feature selection, and none of the papers you mentioned can output the selected features. Thus, we compared our model with unsupervised feature selection methods rather than clustering methods.
> > >
> > > **(2) Reviewer kEpS confounded feature learning and feature selection**
> > >
> > > Feature learning can be divided into feature transformation and feature selection. The first category aims to generate *new* features to better serve the target task, where deep learning and the papers you mentioned belong to this category. On the contrary, feature selection selects the *original* features as proxies of the whole features. Generally speaking, feature transformation achieves better performance, while these new features lose the original semantic for further explanations. Feature selection, although it does not perform as well as feature transformation, has two goals, speed up downstream tasks (Nowadays, it is not so important due to the powerful computational resource) and explanations. It is meaningless to compare feature selection and feature transformation due to their different nature.
> > >
> > > **(3) Reviewer kEpS was not familiar with datasets in the unsupervised feature selection domain**
> > >
> > > Reviewer kEpS quickly checked our competitive methods, including NDFS, TSFS, CAE, and InfFS, and recommended us to use more challenging or common ones. We invite Reviewer kEpS to honestly tell the public whether a common dataset was used in NDFS, TSFS, CAE, and InfFS. The answer is NO—another self-contradiction. On the other hand, among the 12 datasets we used, 10 of them have been adopted by other papers in the feature selection domain, and 2 of them were published in 2018 and have not been explored for the feature selection task.
> > >
> > > Reviewer kEpS suggested using CalTech 101 in InfFS. This point also exposed that Reviewer kEpS was not familiar with unsupervised feature selection. The samples of CalTech 101 in InfFS are formulated with a 4096-dimensional vector extracted from VGG. Remember that the goal of feature selection is to select features for model explanations. Therefore, the original features should have semantic meaning. The VGG features do not. Therefore, CalTech 101 in InfFS is not suitable for feature selection.
> > >
> > > The statements on SSC also suffer from technical issues, which is out of the scope of this paper. We do not point them out here.
> > >
> > > Hope Reviewer kEpS can learn from the above response at the character and knowledge level. Thanks.

---

> > ### Comment · Reviewer_kEpS · 2021-11-25
> > **Towards the limited novelty**
> >
> > As I said earlier, the novelty is not satisfying. The perspective for unsupervised feature selection by feature relationship learning and graph segmentation is not new from my point of view. I will give some evidence in the following.
> >
> > The second-order data in this paper just refers to the correlation/affinity matrix. Let me first briefly summarize the overall pipeline of the proposed method: (1) correlation matrix computation; (2) attention matrix learning based on GCN and contrastive learning; (3) graph construction and segmentation.
> >
> > Recall the subspace-based/spectral clustering methods, such as SSC[1], it also consists of three main steps, including (1) affinity matrix construction, (2) self-representation based affinity matrix optimization, (3) graph embedding and clustering.
> >
> > Compare the proposed method with the subspace-based clustering methods, they have a similar pipeline to process second-order data, and the main differences only lie in the input and affinity matrix optimization. Specifically, the proposed method computes the second-order data on the feature dimension, while SSC and traditional spectral clustering methods construct the affinity matrix on the sample level. The proposed method optimizes the affinity matrix based on the deep learning method, and SSC adopts sparse regularized self-representation. There are also some approaches that learn the affinity matrix based on deep learning, such as SpectralNet [2]. In view of the above comparison, this paper just applies a similar pipeline of spectral clustering to feature selection and presents a GCN-based method to learn a better affinity matrix. Therefore, I believe that the novelty is limited.
> >
> > [1] Sparse subspace clustering: Algorithm, theory, and applications. TPAMI 2013
> > [2] SpectralNet: Spectral Clustering using Deep Neural Networks. ICLR 2018

---

### Official Review · Reviewer_3WQu · 2021-11-02

**Correctness:** 3
**Technical Novelty And Significance:** 3
**Empirical Novelty And Significance:** 3
**Recommendation:** 5
**Confidence:** 4

**Main Review:**

# Strengths

This paper proposed an interesting feature selection algorithm, which leveraged the information from the second-order covariance matrix with the first-order data matrix. On empirical experiments, the proposed method showed superior performance over several baselines, to some extend.

# Weaknesses

Here are some of my comments/questions. I would appreciate it if the author(s) could give a response. If I am wrong, please correct me. Thanks.

- The author(s) stated that "Most of these methods apply the linear feature selection matrices and select the representative features by ranking their feature weight vector. Such operations treat the feature set independently and fail to tackle the complex high-order relationship...". While, as far as I know, the model in [3] actually can globally exploit the complex feature relationships, and the models in [1] and [3] are both flexible to nonlinear feature selection. So what are the limitations of existing methods that the paper was trying to address?

- The authors demonstrated the effectiveness of the proposed method through empirical experiments. Currently, there are many new developments in feature selection, for example, [1] and [2]. I noticed the author(s) cited [1], but why not compare SOFT to more non-linear feature selection models such as AEFS? If so, the effectiveness of the proposed method will be relatively convincing.

- Comparing Table 2 with Figure 3, it is observed that the performance of SOFT is not so stable as other methods, not only on L.Cancer as the author(s) pointed out. So I encourage the author(s) might compare the stability of selected features from SOFT with baseline comparisons (for example, do stability analysis for selected features from different methods) before this paper is published in influential conferences or journals.

- From Table 4 in supplementary material,  it seems that SOFT has no obvious advantage over other baselines in terms of efficiency, doesn't it? If so, efficiency has no advantage and performance is not too stable, then what is the purpose of SOFT? could the author(s) clarify or explain the motivation more? I look forward to hearing that. Thanks.

- Additionally, this paper is generally well written, but some places (some issues) in this paper should be further clarified/fixed.

   - The font for the legend in Figure 3 is too small;

   - The same full name and abbreviation appeared several times in the paper;

   - About the architecture of used baselines, the author(s) stated that "For TSFS, CAE, and InfFS, we use default settings provided in their open-source codes."; however, it is not so clear. For example, for CAE, both linear and nonlinear structures are discussed in [3]. Which one did the author(s) of this paper use?







   - [1] Han et al., Autoencoder inspired unsupervised feature selection. In International Conference on Acoustics, Speech and Signal Processing, 2018.

   - [2] Doquet et al., Agnostic feature selection. In Joint European Conference on Machine Learning and Knowledge Discovery in Databases, 2019.


   - [3] Muhammed Fatih Balın, Abubakar Abid, and James Zou. Concrete autoencoders: Differentiable feature selection and reconstruction. In International conference on machine learning, 2019.

**Summary Of The Paper:**

Feature selection reduces high-dimensional data by identifying comprehensible informative features. This paper proposed an unsupervised feature selection method, SOFT, by combining the information from the second-order covariance matrix with the first-order data matrix. By empirical experiments, the author(s) demonstrated the effectiveness of the proposed method.

**Summary Of The Review:**

In general, I think this paper is relatively interesting, and I was initially interested by the title of this paper. I would like to increase my score if the author(s) could give convincing responses to the previous comments/questions in the Weaknesses part. Thanks.

---

> ### Author Response · Authors · 2021-11-17
> **Response to Reviewer 3WQu (1/3)**
>
> Thanks for your comments, and we appreciate you recognizing the significance of our research problem and soundness of experiments. We are happy to provide extra experiments (Appendix E, F, G, H) and the following response to address all the questions in the Weaknesses part.
>
> ---
>
> **1. Limitations of existing methods that our paper tries to address**
>
> Most existing methods conduct feature selection by ranking the features according to their weights or importance and returning top candidates. As we pointed out in the motivation paragraph in Section 2, the above procedure might lead to redundancy, which disobeys the goal of feature selection. Two features with high similarity are selected or unselected together. In Figure 1, we provided an illustration of feature relations on *Sonar* and *Waveform* datasets by LapScore. Selected features a and b, features c and d on *Sonar* are from two groups within of high similarity, indicating that one feature might be denoted by the other. This drives us to explore the relationships among different features and conduct feature selection by graph segmentation on the feature relationship matrix. Specifically, only one feature is selected from one cluster of the segmentation. By this means, the redundancy of selected features can be alleviated.
>
> ---
>
> **2. Comparison with AEFS [1] and AGNOS [2]**
>
> The reason that we did not provide the results of AEFS and AGNOS is the limited page width. In our previous version, we compared with 10 baseline methods (5 classical ones and 5 recent ones since 2018). If we add more methods, the numbers in Table 2 become very tiny and difficult to read. But here, we would like to report the performance of  AEFS and AGNOS in this response. The reviewer can help judge whether we should put these extra results into the main manuscript.
>
> Table. Accuracy of AEFS, AGNOS, and our method on 12 datasets
>
> |  Dataset | AEFS | AGNOS | Ours |
> |:--------:|:----:|:-----:|:----:|
> |  *COIL20*  | 0.49 |  0.60 | 0.67 |
> |   *Colon*  | 0.54 |  0.54 | 0.56 |
> |  *Gisette* | 0.55 |  0.53 | 0.81 |
> | *L.Cancer* | 0.58 |  0.68 | 0.78 |
> |  *Madelon* | 0.61 |  0.50 | 0.60 |
> | *M.Libras* | 0.38 |  0.38 | 0.46 |
> |   *NCI9*   | 0.33 |  0.41 | 0.44 |
> |    *ORL*   | 0.48 |  0.57 | 0.58 |
> |   *Sonar*  | 0.65 |  0.70 | 0.64 |
> |   *UAV1*   | 0.56 |  0.56 | 0.78 |
> |   *UAV2*   | 0.82 |  0.83 | 0.81 |
> |   *Wave.*  | 0.34 |  0.34 | 0.54 |
> |   Avg.   | 0.53 |  0.55 | 0.64 |
>
> Note that AEFS ran by the author provided codes with the default parameter setting, and AGNOS was implemented by us.
>
> ```
> [1] Han et al., Autoencoder inspired unsupervised feature selection. In International Conference on Acoustics, Speech and Signal Processing, 2018.
> [2] Doquet et al., Agnostic feature selection. In Joint European Conference on Machine Learning and Knowledge Discovery in Databases, 2019.
> ```
>
> ---

---

> > ### Author Response · Authors · 2021-11-17
> > **Response to Reviewer 3WQu (2/3)**
> >
> > ---
> >
> > **3. Stability analysis**
> >
> > We think the reviewer talks about smoothness, rather than stability. Here we first address the smoothness issue and add extra experiments on testing the stability.
> >
> > **(a) Smoothness**. We agree with the reviewer that the performance of SOFT with different percents of selected features is not as smooth as other baseline methods. We conjecture that the phenomena come from the way of feature selection. The existing methods rank the features according to their weights or importance and return top candidates. Thus, 10% selected features are included in the set of 20% selected ones. On the contrary, our feature selection is based on graph segmentation on the learned attention matrix. Due to the non-inheritance of partitions with different cluster numbers, i.e., the numbers of selected features, the partition results might change a lot.
> >
> > To tackle this issue, we followed the suggestion by Reviewer wcmh that a different graph segmentation algorithm could prevent accuracy drop when increasing the selected features and provided extra experiments. The table below shows the standard deviations of SOFT on different percents of selected features without and with multilevel recursive bisection. We can see that more smoothness over different percents is brought by multilevel recursive bisection on *L.Cancer* and *UAV2*, compared with the default one. We invite the reviewer to refer to our new content, Figure 7 in Appendix G, for a better visual experience.
> >
> > Table. Standard deviation over different percents of SOFT default graph segmentation method and multilevel recursive bisection on 12 datasets
> >
> > | Dataset  | Ours(Default) | Our(BiSec)  |
> > |----------|---------------|--------|
> > | *COIL20*   | 0.0120        | 0.0105 |
> > | *Colon*    | 0.0110        | 0.0121 |
> > | *Gisette*  | 0.0479        | 0.0572 |
> > | *L.Cancer* | 0.0957        | 0.0153 |
> > | *Madelon*  | 0.0159        | 0.0216 |
> > | *M.Libras* | 0.0121        | 0.0136 |
> > | *NCI9*     | 0.0085        | 0.0125 |
> > | *ORL*      | 0.0063        | 0.0078 |
> > | *Sonar*    | 0.0312        | 0.0083 |
> > | *UAV1*     | 0.0465        | 0.0018 |
> > | *UAV2*     | 0.1022        | 0.0033 |
> > | *Wave.*    | 0.0157        | 0.0089 |
> > | Avg.     | 0.0337        | 0.0144 |
> >
> > **(b) Stability**. Beyond the above, we also follow the reviewer’s suggestion to do the stability analysis. Please allow us to be long-winded to introduce some basic concepts of stability of feature selection algorithms and make sure we are on the same page. According to [3],  the “stability” of a feature selection algorithm refers to the robustness of its feature preferences with respect to data sampling and to its stochastic nature. An algorithm is ‘unstable’ if a small change in data leads to large changes in the chosen feature subset. Therefore, the stability is different from the performance with various percents of selected features.  A typical approach to measure stability is first to take $M$ bootstrap samples of the provided data set, apply feature selection to each one of them, and then measure the variability in the $M$ feature sets obtained. Since the stability test is timing-consuming due to the bootstrap samples, we conducted the tests on two representative datasets, *COIL20* and *L.Cancer*. We used the stability measurement proposed by [3], ranging from -1 to 1, where the large value means more stability. We generated $M=50$ bootstrap folds. The results in the table below demonstrate our algorithm is much more stable than other baseline methods.  We invite the reviewer to refer to our new content, Figure 6 in Appendix E, for a better visual experience.
> >
> > Table. Stability of different methods with 10% selected features on *COIL20* and *L.Cancer*
> >
> > | Dataset  | LapScore | SPEC   | MCFS   | UDFS   | NDFS   | LRPFS  | NSSLFS | TSFS   | CAE    | InfFS  | Ours   |
> > |----------|----------|--------|--------|--------|--------|--------|--------|--------|--------|--------|--------|
> > | *COIL20*   | 0.5717   | 0.7637 | 0.4215 | 0.0399 | 0.7638 | 0.5185 | 0.0185 | 0.2414 | 0.2532 | 0.8904 | 1.0000 |
> > | *L.Cancer* | 0.3385   | 0.2209 | 0.1807 | 0.1212 | 0.4637 | 0.6777 | 0.2858 | 0.0436 | 0.0506 | 0.2589 | 0.8646 |
> >
> > ```
> > [3] Nogueira, S., Sechidis, K., & Brown, G. (2017). On the stability of feature selection algorithms. J. Mach. Learn. Res., 18(1), 6345-6398.
> > ```
> >
> > ---

---

> > > ### Author Response · Authors · 2021-11-17
> > > **Response to Reviewer 3WQu (3/3)**
> > >
> > > ---
> > >
> > > **4. Efficiency**
> > >
> > > Admittedly the calculation of the second-order feature matrix is time-consuming. When the number of features is much larger than the number of instances, our method runs relatively slow. We guess that SOFT might leave an impression of high computational cost just because of the long execution time on the *Gisette* dataset. Actually, the majority of the running time is occupied by PCA. We followed the suggestion from Reviewer wcmh to try different ways to generate pseudo labels by deep clustering methods. We found the speed of SOFT on the *Gisette* dataset was dramatically enhanced from 15129.18 to 739.55 or 1068.37 by second with DEC [4] or IMSAT [5].  More results can be found in the table below.
> > >
> > > Table. Time cost of our method with different pseudo labeling methods
> > >
> > > | Dataset  | SOFT (Default) | SOFT(DEC) | SOFT(IMSAT) |
> > > |----------|----------------|-----------|-------------|
> > > | *COIL20*   | 180.18         | 22.23     | 36.83       |
> > > | *Colon*    | 46.38          | 40.75     | 55.17       |
> > > | *Gisette*  | 15129.18       | 739.55    | 1068.37     |
> > > | *L.Cancer* | 1.86           | 2.75      | 6.19        |
> > > | *Madelon*  | 75.84          | 18.25     | 35.79       |
> > > | *M.Libras* | 3.80           | 4.17      | 8.03        |
> > > | *NCI9*     | 1034.50        | 6186.08   | 4550.16     |
> > > | *ORL*      | 55.54          | 15.06     | 23.86       |
> > > | *Sonar*    | 2.17           | 3.42      | 6.61        |
> > > | *UAV1*     | 57.40          | 94.37     | 166.51      |
> > > | *UAV2*     | 51.52          | 84.69     | 149.89      |
> > > | *Wave.*    | 15.26          | 25.21     | 44.79       |
> > > | Avg.     | 1387.80        | 603.04    | 512.68      |
> > >
> > > For the motivation of our method, please refer to Q1. If any part of the motivation is unclear to the reviewer, feel free to point it out. We are happy to provide more explanation.
> > >
> > > ```
> > > [4] Xie, J., Girshick, R., & Farhadi, A. (2016, June). Unsupervised deep embedding for clustering analysis. In International conference on machine learning (pp. 478-487). PMLR.
> > > [5] Hu, W., Miyato, T., Tokui, S., Matsumoto, E., & Sugiyama, M. (2017, July). Learning discrete representations via information maximizing self-augmented training. In International conference on machine learning (pp. 1558-1567). PMLR.
> > > ```
> > >
> > > ---
> > >
> > > **5. Other issues**
> > >
> > > Thanks very much for these minor but important points. We would like to seriously address all of them as well.
> > >
> > > a. The font for the legend in Figure 3 has been enlarged.
> > >
> > > b. In the new version, the full name only occurs twice in the abstract and introduction part, respectively.
> > >
> > > c. The default setting of CAE is the non-linear version. We have made this point clear in the new version.
> > >
> > > ---
> > >
> > > Thanks again for helping us improve our paper. All the revised parts, as mentioned in the above point-to-point response, can be found in green in the updated paper. Further interactions are highly welcome, and we are happy to address your comments.

---

> > > > ### Comment · Reviewer_3WQu · 2021-11-24
> > > > **Still not so satisfied with efficiency**
> > > >
> > > > I'm sorry I'm late. I read the author(s)' responses and other reviewers' comments. Thanks to the author(s) for the detailed responses, which clarify most of my questions or doubts, including stability and comparison with other more algorithms.
> > > >
> > > > **1**.For "3. Stability analysis", I denoted the stability analysis of the selected features. As far as I know, the instability of feature selection mainly comes from the randomness of the weights of the learning model and the randomness of the samples used for model learning. The author(s) conducted experiments with the latter, and the results look amazing.
> > > >
> > > > **2**. For "1.Limitations of existing methods that our paper tries to address", as far as I know, the basic idea of CAE is also related to eliminating the redundancy of selected features. But I don't think that it will affect the novelty of the proposed approach in this paper. After all, this paper used a different technique to remove/alleviate redundancy.
> > > >
> > > > **3**. For “4. Efficiency”. For me, I am not satisfied, because feature selection is generally regarded as a dimensionality reduction method for high-dimensional data. If "When the number of features is much larger than the number of instances, our method runs relatively slow", then what is the appeal of the proposed approach?
> > > >
> > > >
> > > > I look forward to the author(s)' further response and please correct me if I am wrong. Thanks in advance.

---

> > > > > ### Author Response · Authors · 2021-11-24
> > > > > **Response to Reviewer 3WQu (Second-round)**
> > > > >
> > > > > Thank you so much for checking our response and recognizing our novelty. We appreciate your extra comments as well. Point 1 and  2 in your comments are correct, where Point 3 is a question that we will respond to below. Specifically, we would like to provide more explanations on the limitations of existing methods and the appeal of our work.
> > > > >
> > > > > ---
> > > > >
> > > > > **1. Limitations of existing methods that our paper tries to address**
> > > > >
> > > > > Generally speaking, feature selection aims to select the representative features as the proxies of all the original features, which can also be understood to remove redundant and irrelevant features. We argue that most unsupervised feature selection methods learn a feature weight vector and rank the features according to their weights. Thank the Reviewer for pointing out CAE, which does not belong to the above category. CAE designs a concrete selector layer based on Concrete random variables and encourages the selector layer to stochastically explore different linear combinations of input features.
> > > > >
> > > > > As the Reviewer pointed out, we use a different technique to alleviate redundancy by graph segmentation on the feature relationship matrix.
> > > > >
> > > > > ---
> > > > >
> > > > > **2. The appeal of our work**
> > > > >
> > > > > As we mentioned in the above point, feature selection aims to remove *redundant* and *irrelevant* features. For the *redundancy* issue, we design the graph segmentation on the feature relationship matrix. On another hand, we design knowledge contrastive distillation to extract the intrinsic feature relationship matrix for *relevant* feature selection. The appeal of our work is that we start from a new perspective for feature selection, i.e., graph segmentation on the feature relationship matrix. To achieve this, we provide a complete framework for feature selection including feature relationship learning and segmentation. We empirically demonstrate our superior and stable performance over numerous classical and recent methods.
> > > > >
> > > > > When handling the datasets with the number of features much larger than the number of instances, our algorithm can still run but relatively slow (we believe every algorithm takes longer time to process large datasets than smaller ones). Actually, we diagnose the most time-consuming part comes from PCA. The unsupervised feature selection task we consider here is not a real-time task, which does not have a high demand on efficiency. Even though the efficiency is not the focus in this paper, we would like to tackle the efficiency issue in future work. We notice there are recent studies [1-3] on accelerating PCA in the literature.
> > > > >
> > > > > ```
> > > > > [1] Gang, Arpita, and Waheed U. Bajwa. "FAST-PCA: A Fast and Exact Algorithm for Distributed Principal Component Analysis." arXiv preprint arXiv:2108.12373 (2021).
> > > > > [2] Feng, Xu, et al. "Fast randomized PCA for sparse data." Asian Conference on Machine Learning. PMLR, 2018.
> > > > > [3] Shamir, Ohad. "Fast stochastic algorithms for SVD and PCA: Convergence properties and convexity." International Conference on Machine Learning. PMLR, 2016.
> > > > > ```
> > > > >
> > > > > ---
> > > > >
> > > > > Thanks again for the guidance to make our paper stronger. We will clarify the above points and our novelty in the final version if our paper is accepted. Happy to have further discussions.

---

### Official Review · Reviewer_ELaK · 2021-11-04

**Correctness:** 3
**Technical Novelty And Significance:** 2
**Empirical Novelty And Significance:** 3
**Recommendation:** 6
**Confidence:** 4

**Main Review:**

strengths
1. The proposed method is interesting;
2. Experimental results are good;
weaknesses
1. Some presentations are not clear enough.
1.1 The first line of page 4, why $\Theta_M$ is forced to be symmetric?
1.2 Why the masked matrix can be defined by Eq.(2)?
2. Since the GCN training process relies on pseudo labels, how to ensure the reliability of the clutering results obtained by PCA and K-means?

**Summary Of The Paper:**

This paper proposes a two-stage second-order unsupervised feature selection via knowledge contrastive distillation model that incorporates the second-order covariance matrix with the first-order data matrix for unsupervised feature selection. A sparse attention matrix that
can represent second-order relations between features is learned in the first stage, a relational graph based on the learned attention matrix is learned to perform graph segmentation for feature selection.

**Summary Of The Review:**

Interesting idea, but some unclear presentations and motivations.

---

> ### Author Response · Authors · 2021-11-16
> **Response to Reviewer ELaK**
>
> Thanks for your comments, and we appreciate you recognizing the significance of our research problem and soundness of experiments. Here we address some clarification questions as follows:
>
> ---
>
> **1. Why is $\theta_M$ symmetric?**
>
> In this paper, we explore the $d\times d$ feature matrix $M_F$, which calculates the similarity between a pair of features. Here we use the inner product, a symmetric metric, to calculate $M_F=X^\top\times X$. Following this, we force the learned attention matrix to be symmetric as well.
>
> ---
>
> **2. Why the masked matrix can be defined by Eq. (2)?**
>
> We assume that the input feature matrix $M_F$ consists of a clean part (attention matrix $M_A$) and a noisy part (masked matrix $M_M$), where the attention matrix is trained to learn the intrinsic data structure. Thus, we define Eq. (2) as a summation formulation. In the revised version, we have added the above assumption and made this point more clear.
>
> ---
>
> **3. Since the GCN training process relies on pseudo labels, how to ensure the reliability of the clustering results obtained by PCA and K-means?**
>
> This is a good question. In the unsupervised feature selection scenario, pseudo labels are used to guide feature selection. In our paper, the attention matrix serving feature selection and pseudo labels can be regarded as a symbiotic relationship. A well-structured attention matrix helps to generate high-quality pseudo labels, and these pseudo labels contribute to better attention matrix learning. In our paper, we jointly optimize both of them in an iterative manner. The table below shows the performance by accuracy of SOFT during running epochs, which demonstrates the effectiveness of guidance from pseudo labels.
>
> Table. Performance by accuracy of SOFT during running epochs
>
> |     Epoch | 0    | 50   | 100  | 150  | 200  | 250  | 300  |
> |--------------|------|------|------|------|------|------|------|
> | *L.Cancer*     | 0.59 | 0.72 | 0.69 | 0.72 | 0.72 | 0.72 | 0.78 |
> | *Madelon*      | 0.57 | 0.59 | 0.54 | 0.50 | 0.56 | 0.55 | 0.60 |
> | *M.Libras*     | 0.44 | 0.44 | 0.44 | 0.46 | 0.44 | 0.47 | 0.46 |
> | *ORL*          | 0.56 | 0.56 | 0.56 | 0.56 | 0.58 | 0.58 | 0.58 |
> | *UAV1*         | 0.56 | 0.56 | 0.78 | 0.78 | 0.56 | 0.56 | 0.78 |
> | *UAV2*         | 0.70 | 0.76 | 0.76 | 0.53 | 0.76 | 0.76 | 0.82 |
>
> ---
>
> Thanks again for helping us improve our paper. All the revised parts, as mentioned in the above point-to-point response, can be found in green in the updated paper. We are open to further interactions, and more comments are welcome.

---

### Official Review · Reviewer_xFNs · 2021-12-02

**Correctness:** 3
**Technical Novelty And Significance:** 3
**Empirical Novelty And Significance:** 3
**Recommendation:** 8
**Confidence:** 4

**Main Review:**

Pros:
1. The motivation is well introduced with illustrative examples in Figure 1.
2. The graph segmentation-based framework is novel and interesting to me, which is different from the mainstream weight-based feature selection.
3. The philosophy of knowledge contrastive distillation is logically sound. I have a minor question that will be posted in the Cons section.
4. The experimental results are extensive. The authors compared with 10 methods including several recent deep methods on 12 datasets and demonstrated the significant improvements.
5. The in-depth exploration is a plus, which helps understand the proposed algorithm.
Cons:
1. The authors employed the knowledge contrastive distillation to learn the feature correlation matrix. The traditional unsupervised feature selection methods based on the first-order feature matrix usually employs a clustering method to purse pseudo labels for feature selection. It is suggested that the authors would like to do another ablation study without knowledge contrastive distillation.
2. The motivation of this paper is to address the redundancy issue of selected features. Although the authors provided the illustrative examples in Figure 1, it would be better to demonstrate the proposed method does not suffer from this issue on the same datasets in the experimental part.


**Summary Of The Paper:**

This paper considers the well-defined feature selection problem. The authors pointed out that the redundancy issue of the weight-based feature selection methods. To tackle this, the authors explored the second-order feature covariance matrix and proposed a two-stage framework including feature correlation matrix via knowledge contrastive distillation and feature selection on the masked correlation matrix via graph segmentation. Extensive experiments demonstrated the effectiveness of the proposed methods.

**Summary Of The Review:**

This paper considers the well-defined feature selection problem. The authors pointed out that the redundancy issue of the weight-based feature selection methods. To tackle this, the authors explored the second-order feature covariance matrix and proposed a two-stage framework including feature correlation matrix via knowledge contrastive distillation and feature selection on the masked correlation matrix via graph segmentation. Extensive experiments demonstrated the effectiveness of the proposed methods.  The motivation is well introduced with illustrative examples. The graph segmentation-based framework is novel and interesting to me, which is different from the mainstream weight-based feature selection. The philosophy of knowledge contrastive distillation is logically sound.
 It is suggested that the authors would like to do another ablation study without knowledge contrastive distillation. Although the authors provided the illustrative examples in Figure 1, it would be better to demonstrate the proposed method does not suffer from this issue on the same datasets in the experimental part.

---

> ### Author Response · Authors · 2021-12-04
> **Response to Reviewer xFNs**
>
> Thanks for your comments, and we appreciate you recognizing the significance of our research problem and the soundness of the experiments. Here we provide more experimental results to address your comments in the Cons section.
>
> ---
>
> **1. Ablation study without knowledge contrastive distillation**
>
> We run our model without the knowledge contrastive distillation part on all the datasets, shown in the Table below (No Distillation). For comparison, we also put the results of Fig.(b) in this Table. Here ‘First Order’ denotes a model using original feature instead of feature relation matrix as input and adopts a vectorial learnable mask for knowledge contrastive distillation and feature ranking. ‘Covariance’ denotes using the original feature matrix directly for the graph segmentation stage. We can see that our proposed method performs the best on average among these methods. Noticing that ‘No Distillation’ method learns a relation matrix, but its performance is not better than ‘Covariance.’ We analyze that ‘No Distillation’ employs a clustering method to pursue pseudo labels for feature selection, which might suffer from noises in the original features when generating pseudo labels and thus hurt the learning process. While our method introduces the knowledge contrastive distillation part, it helps in removing noises from original features for learning a better relation matrix. We will add more descriptions on the contribution of the knowledge contrastive distillation part in our paper.
>
> Table. Ablation Study of the proposed method
>
> | Dataset        | First Order | Covariance | No Distillation  | Ours |
> |----------------|-------------|------------|------------------|------|
> | *COIL20*         | 0.67        | 0.68       | 0.68             | 0.67 |
> | *Colon*          | 0.52        | 0.53       | 0.56             | 0.56 |
> | *Gisette*        | 0.70        | 0.71       | 0.71             | 0.81 |
> | *L.Cancer*    | 0.62        | 0.78       | 0.68             | 0.78 |
> | *Madelon*        | 0.51        | 0.55       | 0.58             | 0.6  |
> | *M.Libras* | 0.40        | 0.46       | 0.44             | 0.46 |
> | *NCI9*           | 0.43        | 0.46       | 0.44             | 0.44 |
> | *ORL*            | 0.55        | 0.56       | 0.56             | 0.58 |
> | *Sonar*          | 0.54        | 0.64       | 0.51             | 0.64 |
> | *UAV1*           | 0.55        | 0.78       | 0.66             | 0.78 |
> | *UAV2*           | 0.53        | 0.83       | 0.75             | 0.81 |
> | *Wave.*  | 0.48        | 0.50       | 0.50             | 0.54 |
> | Avg.           | 0.54        | 0.62       | 0.59             | 0.64 |
>
> ---
>
> **2. Demonstration that the proposed method does not suffer from the redundancy issue on the same datasets with the illustrative examples**
>
> We calculate the maximum pairwise correlation of selected features on the illustrative example datasets, *Sonar* and *Waveform*. LapScore gets 1.00 and 1.00 on both datasets, while our method gets 0.47 and 0.17, respectively. This indicates that our method can alleviate the redundancy issue among the selected features. We would like to provide a figure to better demonstrate this point in the next version.
>
> ---
>
> Thanks again for helping us improve our paper. We are open to further interactions, and more comments are welcome.

---

### Author Response · Authors · 2021-11-17
**General Response and Paper Update**

We would like to thank the reviewers for all the valuable and constructive reviews! We take every comment seriously and have added new experiments in Appendix E, F, G, H:

1. Analysing stabilities of different methods on two datasets to show the robustness of our method;

2. Extending experiments by using different pseudo labeling methods, which significantly accelerates our method;

3. Extending experiments by adopting another graph segmentation method to help explain and address the accuracy drop issue when the number of selected features increases;

4. Doing an ablation study of our method on noise removal to show the effect of different noise removal strategies.

All the revisions are colored in green in the updated manuscript. We are open to further interactions, and more comments are welcome. Thank you.

---

### Decision · Program_Chairs · 2022-01-20

**Decision:**

Reject

**Comment:**

This paper proposes a new two stage second-order unsupervised feature selection method via knowledge contrastive distillation. In the first stage, a sparse attention matrix that represents second order statistics is learned. In the second stage, a relational graph based on the learned attention matrix is constructed to perform graph segmentation for feature selection.

This proposed method contains some new and interesting ideas and is novel in the unsupervised feature selection setting, though some components such as the second order affinity matrix are not totally new. The proposed method is technically sound. The authors compared their method with 10 methods including several recent deep methods on 12 datasets and demonstrated consistent improvements.

However, there are some concerns from the reviewers, even after the discussion phase. 1) The computational efficiency of the proposed method seems to be low. Since one goal of feature selection is to speed up downstream tasks, the efficiency of feature selection itself should also be considered. I suggest the authors analyze the computational bottleneck of the proposed method and improve the efficiency. 2) More ablation studies can be added to illustrate how the proposed method removes the redundancy issues of the selected features. 3) Some metrics like supervised classification accuracy can be potentially used as a metric. Though supervised classification is impossible in the unsupervised learning setting, running the experiments on some datasets that have labels by pretending having on label is one way to evaluate the method.

Overall, the paper provides some new and interesting ideas. However, given the above concerns, the novelty and significance of the paper will degenerate. Although we think the paper is not ready for ICLR in this round, we believe that the paper would be a strong one if the concerns can be well addressed.